# Aurkb/PP1-mediated resetting of Oct4 during the cell cycle determines the identity of embryonic stem cells

Jihoon Shin[1†], Tae Wan Kim[1†], Hyunsoo Kim[1†], Hye Ji Kim[2†], Min Young Suh[3], Sangho Lee[3], Han-Teo Lee[4], Sojung Kwak[1], Sang-Eun Lee[1,5], Jong-Hyuk Lee[1], Hyonchol Jang[6], Eun-Jung Cho[7], Hong-Duk Youn[1,3,4*]

[1]National Creative Research Center for Epigenome Reprogramming Network, Department of Biomedical Sciences, Ischemic/Hypoxic Disease Institute, Seoul National University College of Medicine, Seoul, Republic of Korea; [2]Department of Biological Sciences, Seoul National University, Seoul, Republic of Korea; [3]Department of Molecular Medicine and Biopharmaceutical Sciences, Graduate School of Convergence Science, Seoul National University, Seoul, Republic of Korea; [4]Interdisciplinary Program in Genetic Engineering, Seoul National University, Seoul, Republic of Korea; [5]Department of Internal Medicine, Seoul National University Hospital, Seoul, Republic of Korea; [6]Division of Cancer Biology, Research Institute, National Cancer Center, Goyang, Republic of Korea; [7]College of Pharmacy, Sungkyunkwan University, Suwon, Republic of Korea

*For correspondence: hdyoun@snu.ac.kr

[†]These authors contributed equally to this work

Competing interests: The authors declare that no competing interests exist.

**Abstract** Pluripotency transcription programs by core transcription factors (CTFs) might be reset during M/G1 transition to maintain the pluripotency of embryonic stem cells (ESCs). However, little is known about how CTFs are governed during cell cycle progression. Here, we demonstrate that the regulation of Oct4 by Aurora kinase b (Aurkb)/protein phosphatase 1 (PP1) during the cell cycle is important for resetting Oct4 to pluripotency and cell cycle genes in determining the identity of ESCs. Aurkb phosphorylates Oct4(S229) during G2/M phase, leading to the dissociation of Oct4 from chromatin, whereas PP1 binds Oct4 and dephosphorylates Oct4(S229) during M/G1 transition, which resets Oct4-driven transcription for pluripotency and the cell cycle. Aurkb phosphor-mimetic and PP1 binding-deficient mutations in Oct4 alter the cell cycle, effect the loss of pluripotency in ESCs, and decrease the efficiency of somatic cell reprogramming. Our findings provide evidence that the cell cycle is linked directly to pluripotency programs in ESCs.

## Introduction

Embryonic stem cells (ESCs) have unique transcriptional programs for self-renewal and pluripotency which differentiates into all types of cells. Core transcription factors—Oct4, Sox2, Nanog (OSN)—govern such pluripotency transcriptional programs (*Jaenisch and Young, 2008*; *Young, 2011*). ESCs grow rapidly and undergo an unusual cell cycle, characterized by a very short G1 phase and a long S phase in mouse and human (*Kapinas et al., 2013*; *Savatier et al., 1994*; *White and Dalton, 2005*). The duration of G1 in mouse ESCs and human ESCs determines their fate with regard to differentiation and pluripotency (*Coronado et al., 2013*; *Mummery et al., 1987*; *Pauklin and Vallier, 2013*). A recent study revealed that the S and G2 phases tend to maintain the pluripotent state at early time of differentiation (*Gonzales et al., 2015*). Thus, cell cycle regulation in ESCs should be linked to pluripotency in maintaining ESC identity.

**eLife digest** Embryonic stem cells can give rise to any type of cell in the body – an ability known as pluripotency. These cells rapidly divide and self-renew until they are exposed to signals that cause them to mature into a particular specialized cell type.

As cells prepare to divide, they transition through a series of phases known as the cell cycle. In embryonic stem cells, these phases are often shorter than in other cell types. This altered timing is thought to be important for maintaining the pluripotency of the stem cells.

Proteins called core transcription factors also help stem cells to remain pluripotent. Evidence suggests that the activity of some of these proteins affects the timing of the different cell cycle phases. However, it is not clear exactly how they do so or how the activity of the transcription factors is controlled.

A core transcription factor called Oct4 is thought to be a "master regulator" of pluripotency that controls the activity of many of the other core transcription factors. Shin, Kim, Kim, Kim et al. have now studied the activity of Oct4 around the point of cell division. This revealed that a protein called aurora kinase B modifies Oct4 by adding a phosphate group to it just before a cell divides. This modification causes Oct4 to detach from chromatin, the protein structure in which DNA is packaged inside cells.

Following cell division, another protein called PP1 removes the phosphate group from Oct4. This "resets" the pluripotency of the stem cell, allowing it to continue to self-renew. Cells that contain only mutant forms of Oct4 that cannot bind to aurora kinase B or PP1 lose their pluripotency. The mutant Oct4 proteins also alter the cell cycle of the stem cells.

Overall, Shin et al.'s findings suggest that Oct4 regulates the cell cycle of embryonic stem cells as well as their pluripotency. How Oct4 activity affects the specialization of the stem cells into mature cell types remains to be investigated in future studies.

Our understanding of the molecular associations between the cell cycle and pluripotency in ESCs is limited. Robust Cdk2 activity shortens G1 phase by inducing rapid G1-S transition and promotes pluripotency and self-renewal in ESCs (*Neganova et al., 2009*; *Van Hoof et al., 2009*). Nanog controls S phase entry by targeting Cdc25c and Cdk6 (*Zhang et al., 2009*). Despite the growing evidence on the direct connection between the cell cycle and pluripotency, it remains unknown how ESCs preserve their pluripotency through cell cycle progression and reset pluripotency transcriptional programs during the transition from mitosis to G1 phase.

Oct4 is considered a master regulator of ESC pluripotency through its cooperation with other core transcription factors (*Jerabek et al., 2014*). Post-translational modifications to Oct4 affect its transcriptional activity and lead to ESC pluripotency. For example, we previously reported that O-GlcNAcylatoin of murine Oct4(T228) is important for ESC pluripotency and somatic cell reprogramming (*Jang et al., 2012*). Also, Oct4 is controlled by phosphorylation (*Brumbaugh et al., 2012*; *Saxe et al., 2009*; *Spelat et al., 2012*), but there is no evidence that phosphorylation-mediated regulation of Oct4 during the cell cycle affects Oct4-mediated pluripotency programs in ESCs.

The Aurora kinase b (Aurkb)-protein phosphatase 1 (PP1) axis is critical for kinetochore assembly/disassembly during the cell cycle, regulating the balance between phosphorylation and dephosphorylation of kinetochore substrates (*Emanuele et al., 2008*; *Kim et al., 2010*). Specifically, PP1 mediates the M/G1 transition and ensures proper resetting of the subsequent G1 phase by dephosphorylating cell cycle machinery (*Ceulemans and Bollen, 2004*). When the cell cycle resets, transcriptional programs for ESC pluripotency should be reset, because transcriptional programs are generally switched off at the onset of mitosis and subsequently reestablished during entry into the next G1 phase (*Delcuve et al., 2008*; *Egli et al., 2008*; *Martinez-Balbas et al., 1995*). Thus, during the cell cycle, the Aurkb-PP1 axis might be linked directly to the post-translational modification of pluripotency factors with regard to the resetting of pluripotency in ESCs.

In this study, we demonstrate that the Aurkb-PP1 axis regulates Oct4 during the cell cycle over time and by location. We found that Oct4 contains a well-conserved Aurkb phosphorylation residue (S229) and PP1 binding motif (RVXF) in its homeodomain. Aurkb phosphorylates Oct4(S229) during

G2/M phase and dissociates p-Oct4(S229) from chromatin, and PP1 dephosphorylates p-Oct4(S229) during the M/G1 transition, which prompts Oct4 to reset pluripotency transcription on re-entry into the following G1 phase. We found that mutating the Aurkb-phosphorylation residue S229 and the PP1-binding residue F271 of Oct4 in ESCs led to a significant loss of pluripotency and altered the cell cycle. Transduction of these mutants into MEFs significantly decreased the reprogramming efficiency.

Based on these findings, we propose that the spatiotemporal regulation of Oct4 by the Aurkb-PP1 axis during the cell cycle is critical for resetting pluripotency and cell cycle genes in determining the identity of ESCs.

## Results

### Phosphorylated Oct4 at serine 229 is highly enriched in G2/M phase and dissociated from chromatin

To understand the function of the phosphorylation of Oct4, we examined its phosphorylation sites by transient transfection of Flag-Oct4 into E14 ESCs, analyzed the phosphorylation state of purified Oct4 by mass spectrometry, and identified 4 phosphorylation sites (*Figure 1—figure supplement 1A and B*). We then generated phosphor-mimetic mutants and measured their transcriptional activities by transfecting them into NIH-3T3 cells that stably harbored Oct4-driven luciferase reporter genes (*Figure 1—figure supplement 1C*). Only the S229D mutant significantly reduced Oct4 transcriptional activity. Notably, serine 229 lies in the N-terminal region of the homeodomain of Oct4 and is well conserved throughout many species (*Figure 1—figure supplement 1D and E*).

Next, we generated a rabbit polyclonal antibody against phosphorylated Oct4(S229) [thereafter p-Oct4(S229)] and confirmed its specificity by dot blot and western blot (*Figure 1—figure supplement 2A and B*).

We then examined p-Oct4(S229) expression by confocal microscopy in undifferentiated E14 ESCs (*Figure 1A* upper panel). p-Oct4(S229) in E14 ESCs was detected locally around mitotic cells. From this result above, we wondered whether Oct4 phosphorylation at serine 229 occurs in a cell cycle-dependent manner.

To this end, we treated cells with various agents that are related to the cell cycle and DNA damage. Notably, treatment of ESCs with nocodazole significantly enhanced Oct4 phosphorylation at S229, but aphidicolin and adriamycin decreased p-Oct4(S229) levels (*Figure 1—figure supplement 2C–E* and *Figure 1B*). In previously published data, 18 hr incubation with 200ng/ml of nocodazole was enough to synchronize hESCs at G2/M phase without inducing differentiation (*Zhang et al., 2009*). In the case of E14 ESCs, nocodazole treatment during 10 hrs completely arrested cells at G2/M phase. On treatment with nocodazole, p-Oct4(S229) began to rise in late S phase or early G2/M phase, peaking at G2/M phase (*Figure 1A–C*).

To confirm the localization of p-Oct4(S229), we adapted the fluorescence ubiquitination cell cycle indicator (FUCCI) reporter system to ESCs (*Sakaue-Sawano et al., 2008*). We generated E14 ESCs that stably expressed GFP-mAG-geminin during S-G2-M phase and examined p-Oct4(S229) by confocal microscopy on nocodazole treatment (*Figure 1D*). As expected, p-Oct4(S229) overlapped with GFP-geminin in G2/M phase. In contrast, p-Oct4(S229) did not merge with Red-mKO2-Cdt1, which is highly expressed in G1 phase (*Figure 1E*).

We then confirmed that Oct4 dissociates from the binding region of Oct4 and Nanog at G2/M phase by ChIP-qPCR (*Figure 1F*). Under the same conditions, p-Oct4(S229) rarely bound to the same locus, despite p-Oct4(S229) was successfully pulled down with the antibody (*Figure 1—figure supplement 2F*). This result is consistent with a previous report that a human phosphor-mimetic form of Oct4(S325E) [homolog of mouse Oct4(S229)] binds to DNA more weakly than Oct4(WT) by in vitro EMSA (*Brumbaugh et al., 2012*). The loss of DNA binding affinity of Oct4 by phosphorylation might be induced by steric and electrostatic clashes (*Saxe et al., 2009*). Based on these findings, Oct4 is specifically phosphorylated at serine 229, and p-Oct4(S229) dissociates from chromatin in G2/M phase.

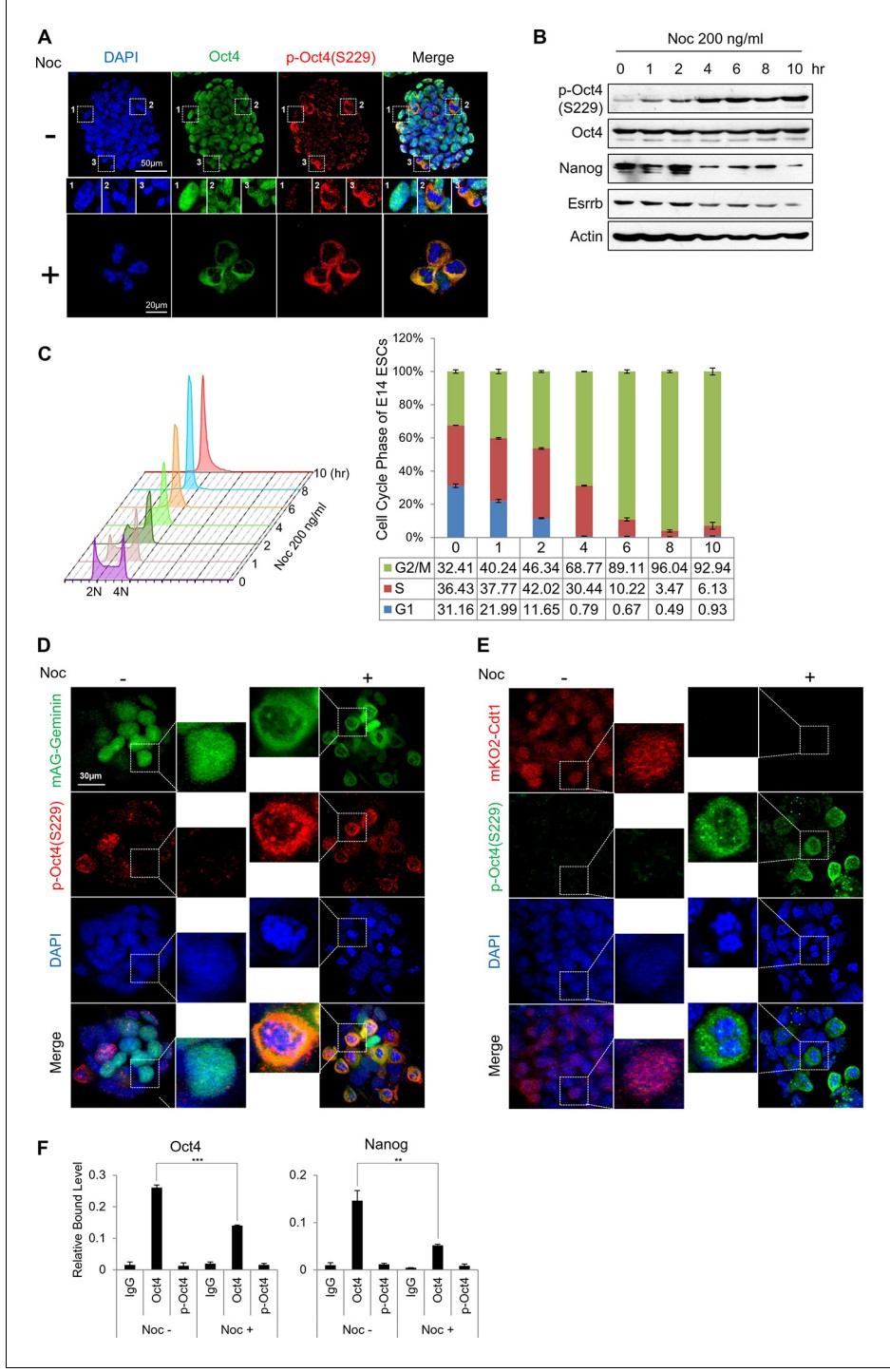

**Figure 1.** Phosphorylated Oct4 at serine 229 is enriched in G2/M phase and dissociated from chromatin. (**A**) Immunostaining of E14 ESCs treated with or without nocodazole (NOC, 200 ng/ml) for 10 hr. Oct4 was stained with anti-Oct4 (green), p-Oct4(S229) was stained with anti-p-Oct4(S229) (red), and DNA was stained with DAPI (blue). White boxes represent cells at various stages. Shown are interphase (1), metaphase (2), and anaphase (3) cells. Scale bars were shown. (**B**) E14 ESCs were treated with nocodazole (200 ng/ml) for the indicated times and immunoblotted with the indicated antibodies. Phosphorylation levels of Oct4 at serine 229 were gradually induced during nocodazole treatment. (**C**) Histograms of the proportions of nocodazole-treated (200 ng/ml) E14 ESCs at various stages in the cell cycle. Cells were stained with PI and DNA contents were analyzed by FACS ($1 \times 10^4$ cells/sample). (**D** and **E**) Fluorescence images of E14 ESCs expressing mKO2-Cdt1 and mAG-Geminin (FUCCI reporter). Shown are green (mAG-geminin) and red (mKO2-Cdt1) fluorescence. E14 ESCs expressing FUCCI reporter were

*Figure 1 continued on next page*

*Figure 1 continued*

left untreated or treated with nocodazole (NOC, 200 ng/ml) for 10 hr. p-Oct4(S229) was stained with anti-p-Oct4 (S229) (red, *Figure 1E*; green, *Figure 1F*), and DNA was stained with DAPI (blue). Scale bars, 30 μm (**F**) ChIP-qPCR assay was performed with anti-IgG, anti-Oct4, and anti-p-Oct4(S229) in E14 ESCs with or without nocodazole (NOC, 200 ng/ml) for 10 hr. Values represent mean ± standard deviation (n≥3). (**p<0.01, ***p<0.001)

The following figure supplements are available for figure 1:

**Figure supplement 1.** Identification of Oct4 phosphorylation at serine 229 residue.

**Figure supplement 2.** Characterization of Oct4 phosphorylation site at serine 229 and anti-p-Oct4(S229) antibody.

## Aurora kinase b binds Oct4 and phosphorylates Oct4 at serine 229 in a cell cycle dependent manner

To identify the kinases that phosphorylate Oct4(S229), we selected 19 candidates using a group-based prediction system (*Xue et al., 2008*) and among Oct4-interacting kinases (*Ding et al., 2012*) (*Figure 2—figure supplement 1A*). We examined the phosphorylation of S229 by these 19 recombinant kinases by in vitro kinase assay and western blot with anti-pOct4(S229)—6 kinases could phosphorylate S229 (*Figure 2—figure supplement 1B*).

To identify the kinases that mediate Oct4(S229) phosphorylation during G2/M phase, we screened the kinases by administering nocodazole to ESCs that were knocked down with cognate lentivirally expressed shRNAs of kinases (*Figure 2—figure supplement 1C and D*). p-Oct4(S229) levels declined significantly on knockdown of *Aurkb*. Further, we confirmed that recombinant Aurkb phosphorylated GST-Oct4(S229) by in vitro $^{32}$P-ATP-labeled kinase assay and western blot with anti-p-Oct4(229) (*Figure 2A and B*).

To verify the Aurkb-mediated phosphorylation of Oct4(S229), we treated nocodazole-pretreated E14 ESCs (10 hr) with various aurora kinase inhibitors for 15 min. An Aurkb-specific inhibitor, hesperadin, completely blocked the phosphorylation, but an Aurka-specific inhibitor, MLN8237, did not. AT9283, an inhibitor of both Aurka and Aurkb, prevented phosphorylation (*Figure 2C*). Under this condition, Aurkb inhibition did not alter cell cycle profile (*Figure 2D*). Aurkb preferentially phosphorylates serine when arginine lies 2 residue upstream of a phosphoserine (-2 position) (*Sugiyama et al., 2002*). In Oct4, we found arginine-227, residing 2 residues upstream of S229 (*Figure 1—figure supplement 1E*).

We then observed that Flag-Aurkb interacts with endogenous Oct4 in E14 ESCs by immunoprecipitation (*Figure 2E*). To determine the cell cycle phases during which Oct4 preferentially interacts with Aurkb, Flag-Oct4-expressing ZHBTc4 ESCs were pretreated with nocodazole for 6 hr, maintaining them in G2/M phase, and released on removal of nocodazole for the cell cycle progression. Notably, Flag-Oct4 interacted strongly with endogenous Aurkb in G2/M phase in Flag-Oct4-expressing ZHBTc4 ESCs (*Figure 2F and G*), consistent with our result that Oct4(S229) is heavily phosphorylated in G2/M phase (*Figure 1*). These findings demonstrate that Aurkb is the kinase that phosphorylates Oct4(S229) in G2/M phase.

## Protein phosphatase 1 binds Oct4 and dephosphorylates serine 229 in Oct4 in G1 phase

When nocodazole treated ZHBTc4 ESCs were released into normal serum, the Aurkb-Oct4 interaction weakened and p-Oct4(S229) levels declined (*Figure 2F*), indicating that certain phosphatases catalyze the dephosphorylation of p-Oct4(S229) during the M/G1 transition.

In examining the amino acid sequence of Oct4, we found that it contains a protein phosphatase 1 (PP1)-binding sequence (268-RVWF-271) in its homeodomain, near the S229 Aurkb phosphorylation site in the 3-dimensional structure (*Figure 3A and B*). This motif is well conserved among many species (*Figure 3—figure supplement 1A*). Thus, we studied the interaction of Oct4 with 3 isoforms of PP1: PP1α, PP1β, and PP1γ. We found that Oct4 interacted more strongly with endogenous PP1β and PP1γ than with PP1α in ZHBTc4 ESCs (*Figure 3C*).

Next, we examined the interaction between Oct4 and PP1 isoforms during the M/G1 transition after nocodazole treatment and release into normal serum (*Figure 3D*). p-Oct4(S229) disappeared

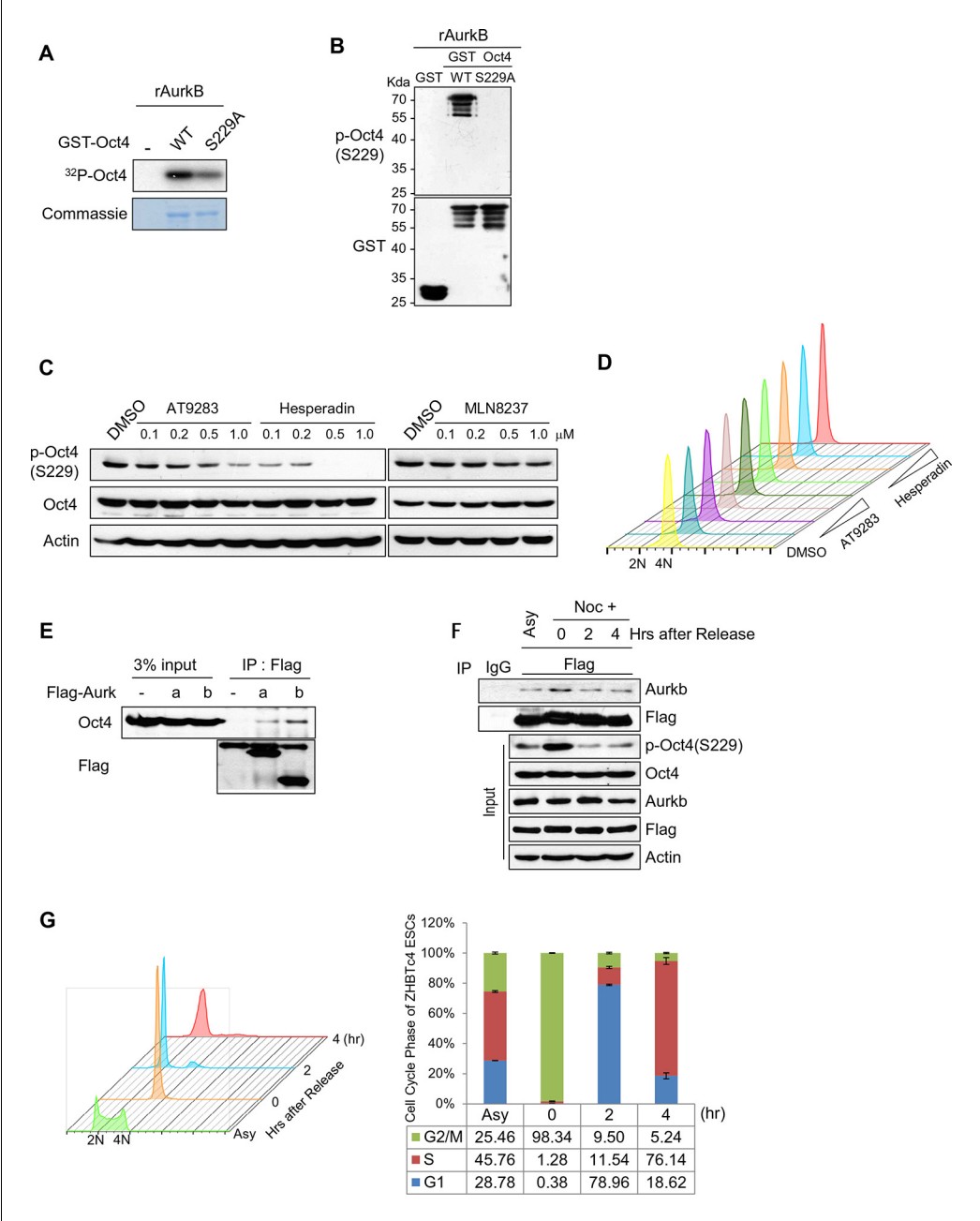

**Figure 2.** Aurkb binds and phosphorylates Oct4 at serine 229 during G2/M phase. (**A**) Radioactive in vitro kinase assay using recombinant Aurkb to phosphorylate GST-Oct4 WT and S229A mutant. Coomassie staining of purified proteins and autoradiogram showing incorporation of γ-$^{32}$P ATP. (**B**) Cold in vitro kinase assay reactions using recombinant Aurkb with purified GST, GST-Oct4 WT, and S229A mutant as substrate followed by western blot. (**C** and **D**) Nocodazole-arrested E14 ESCs (200 ng/ml for 10 hr) were treated with the Aurora kinase inhibitors AT9283 (inhibits Aurka and Aurkb), hesperadin (inhibits Aurkb), and MLN8237 (inhibits AurkA). Gradual decreases in p-Oct4(S229) levels with increasing concentrations of Aurkb inhibitors in E14 ESCs were seen by western blot (**C**). FACS analysis was performed under the same condition (**D**) (1x10$^4$ cells/sample). (**E**) Coimmunoprecipitation of Oct4 with Aurka and Aurkb from E14 ESCs stably expressing Flag-tagged Aurora kinases. (**F**) Changes in Oct4 interaction with Aurkb during cell cycle progression. Whole-cell lysates from Flag-Oct4-expressing ZHBTc4 ESCs were pulled down with anti-Flag beads. Bound proteins were immunoblotted with the indicated antibodies. (**G**) DNA content analysis of Flag-Oct4 expressing ZHBTc4 ESCs by FACS. Flag-Oct4-expressing ZHBTc4 ESCs, treated with nocodazole (200 ng/ml) for 6 hr, were released for 2 and 4 hr and DNA contents were counted (1x10$^4$ cells/sample).

The following figure supplement is available for figure 2:

**Figure supplement 1.** Screening of kinases responsible for phosphorylation on Oct4 at S229.

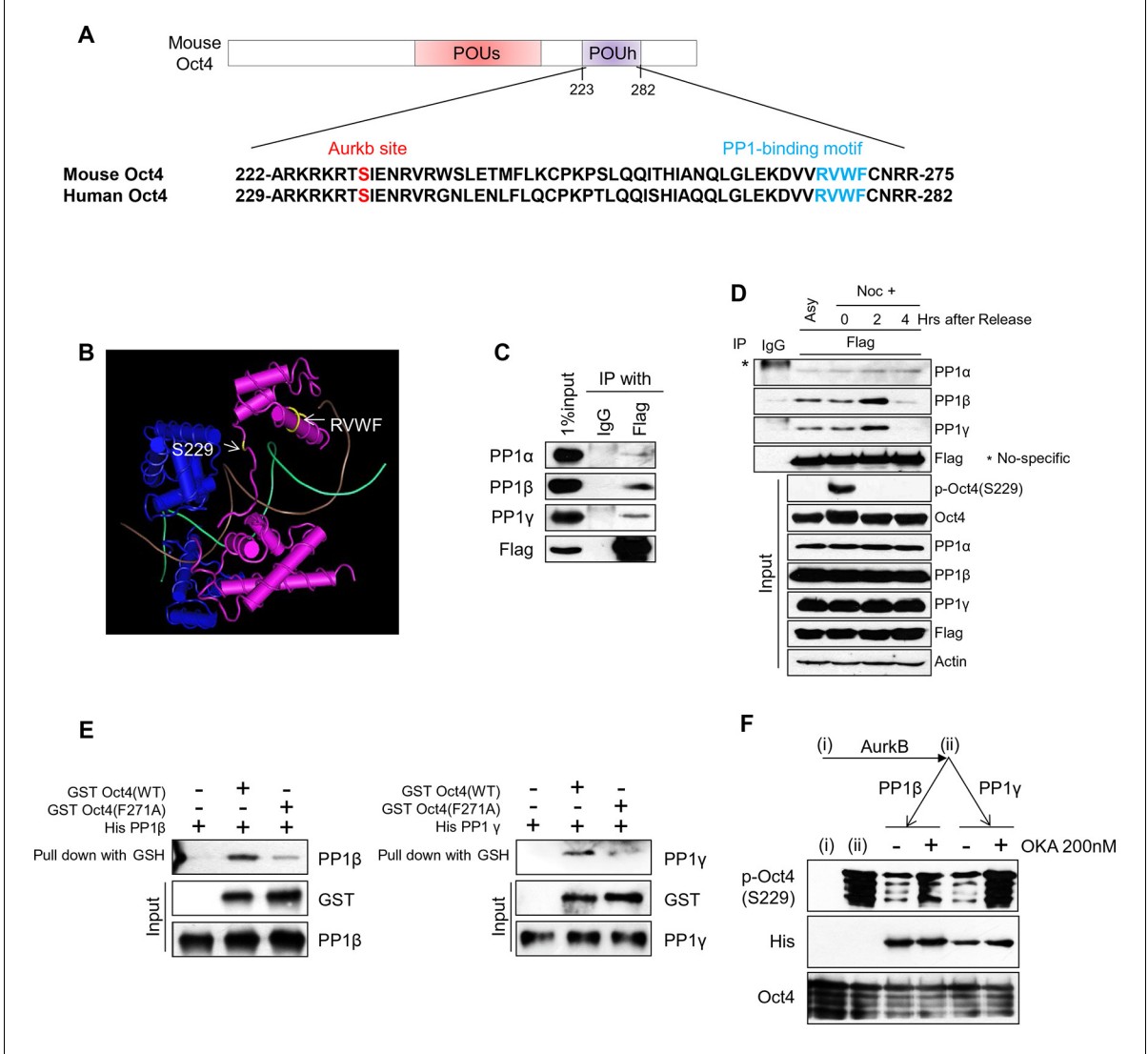

**Figure 3.** PP1 binds and dephosphorylates Oct4 at serine 229 during G1 phase. (**A**) Sequence alignment of Oct4. Oct4 contains a conserved PP1 docking motif (RVXF). (**B**) Three-dimensional structure of Oct4 and DNA complex (MMDB ID: 87311) was adapted from the Molecular Modeling Database (MMDB) of NCBI. Each yellow region indicates S229 and an RVWF PP1-binding domain. (**C**) Coimmunoprecipitation assay revealing the endogenous interaction between Oct4 and PP1 catalytic subunits. Proteins were immunoprecipitated from Flag-Oct4-expressing ZHBTc4 ESCs with Flag antibody, followed by western blot. (**D**) Changes in Oct4 interaction with PP1 catalytic subunits during cell cycle progression. Whole-cell lysates from Flag-Oct4-expressing ZHBTc4 ESCs were pulled down with anti-Flag beads. Immunoprecipitated proteins were immunoblotted with the indicated antibodies. (**E**) Purified GST-Oct4(WT) or GST-Oct4(F271A) mutant was incubated with purified (His)$_6$-PP1β and PP1γ and then pulled down with GST beads. Immunoblot shows that PP1β and γ directly bind GST-Oct4(WT). PP1β and PP1γ show weaker interaction with GST-Oct4(F271A) than wild-type Oct4. (**F**) In vitro phosphatase assay using PP1β or PP1γ with phosphorylated Oct4 as substrate. Okadaic acid (OKA) treatment decreased PP1-mediated dephosphorylation of Oct4.

The following figure supplement is available for figure 3:

**Figure supplement 1.** PP1 dephosphorylates Oct4 at S229 in vitro and in vivo .

quickly during the M/G1 transition. In parallel, PP1β and PP1γ interacted strongly with Oct4 in G1 phase (2 hr after release). However, PP1α bound weakly to Oct4 regardless of cell cycle stage, suggesting that PP1α might be not a true p-Oct4(S229) phosphatase during the M/G1 transition. Thus, we focused on PP1β and PP1γ with regard to the dephosphorylation of Oct4 in subsequent experiments.

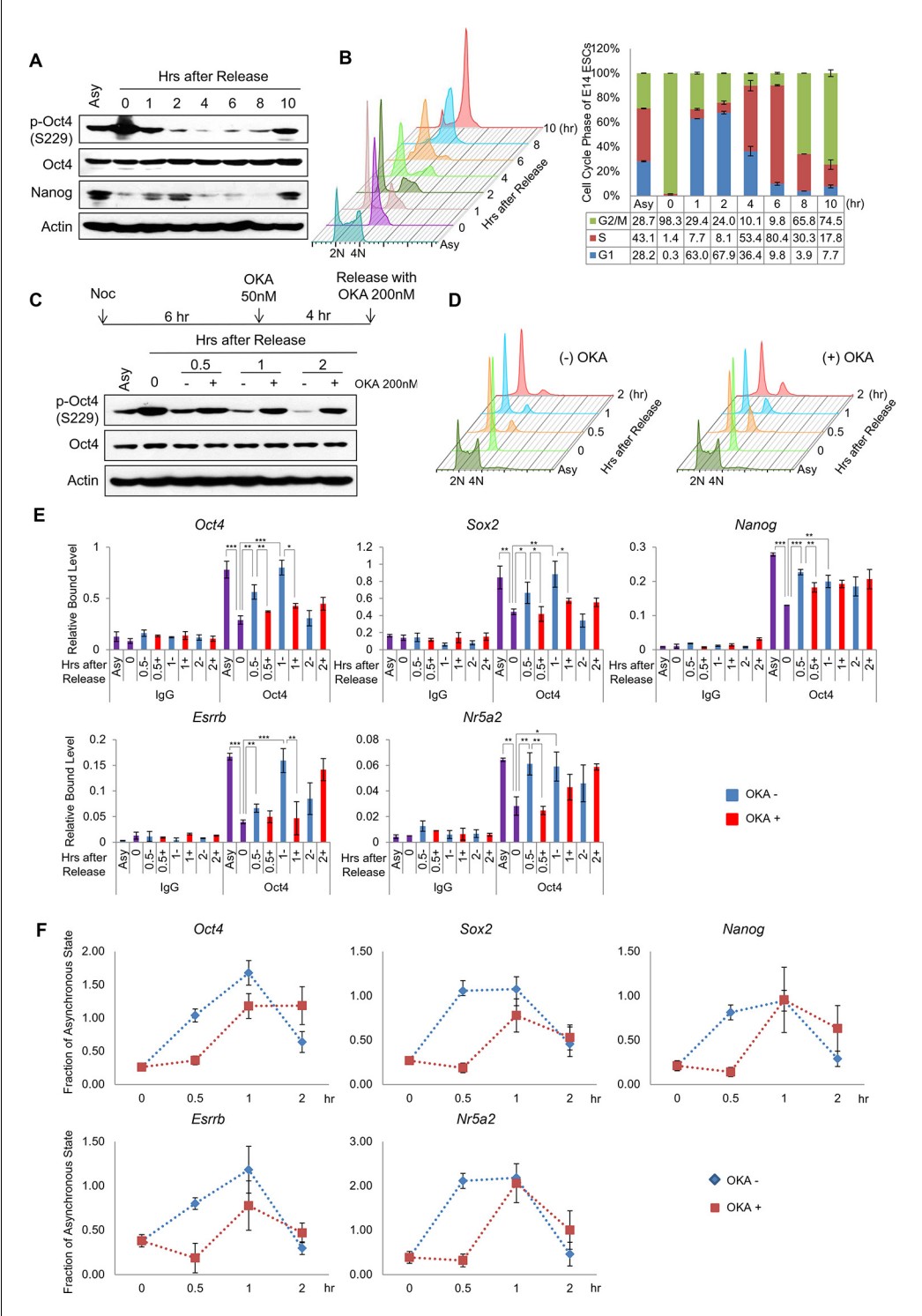

**Figure 4.** PP1-mediated dephosphorylation of Oct4(S229) correlates with the resetting of pluripotency genes in the next G1 phase. (A) p-Oct4(S229) levels after release of E14 ESCs from M-phase arrest. Shown are immunoblots for the indicated proteins. (B) Nocodazole-treated E14 ESCs were released and analyzed for DNA content by FACS (1x10⁴ cells/sample). (C) OKA treatment retards dephosphorylation of p-Oct4(S229) during the M/G1 phase transition. The experimental strategy is shown (upper panel). The same strategy was applied to (D–F). Whole-cell lysates from E14 ESCs were collected and assessed by western blot. (D) Histogram shows cell cycle state of E14 ESCs without (upper panel) or with (lower panel) OKA treatment. (E) ChIP-qPCR analysis of E14 ESCs with anti-Oct4 in regions of pluripotency-associated Oct4 target genes during the M/G1 phase transition with or without OKA treatment. IgG was used as a control. Values represent mean ± standard deviation (n≥3). t-test was used to calculate the statistical significance of differences in enrichment levels of Oct4 at pluripotency-associated Oct4 target genes in ESCs during the M/G1 transition with or without

*Figure 4 continued on next page*

*Figure 4 continued*

OKA. (*p<0.05, **p<0.01, ***p<0.001) (**F**) Nascent RNA of pluripotency-associated Oct4 target genes from E14 ESCs were collected and analyzed by real-time qPCR during the M/G1 phase transition with or without OKA. Levels of each nascent RNA were normalized by those in asynchronous E14 ESCs.

The following figure supplement is available for figure 4:

**Figure supplement 1.** Dissociation of p-Oct4(S229) from chromatin occurs independent to chromatin status.

We altered phenylalanine-271 to alanine in Oct4 [Oct4(F271A)] by site-directed mutagenesis and measured the in vitro interaction between PP1 and Oct4(WT) or Oct4(F271A) using bacterially purified recombinant (His)$_6$-PP1β, (His)$_6$-PP1γ, GST-Oct4(WT), and GST-Oct4(F271A). (His)$_6$-PP1β and γ bound more robustly to GST-Oct4(WT) but weakly to GST-Oct4(F271A) (*Figure 3E*), indicating that Oct4 interacts directly with PP1 through a PP1-binding motif (RVWF).

We then determined whether PP1 dephosphorylates the Aurkb-catalyzed p-Oct4(S229) by preincubating recombinant GST-Oct4 with recombinant Aurkb, adding purified PP1, and measuring the phosphorylation state of p-Oct4(S229) by western blot. Recombinant PP1β and PP1γ, but not PP1α, dephosphorylated Aurkb-mediated phospho-Oct4(S229) (*Figure 3—figure supplement 1B*). Further, pretreatment with okadaic aid (OKA), a PP1 inhibitor, blocked the dephosphorylation of p-Oct4(S229) (*Figure 3F*), indicating that the interaction between Oct4 and PP1 is important for dephosphorylation of phospho-S229 in Oct4. Treatment of E14 ESCs with 50 nM OKA increased p-Oct4(S229) levels after 4 hr (*Figure 3—figure supplement 1C*). Rising concentrations of OKA (from 0–200 nM) gradually increased p-Oct4(S229) levels after 2 hr of treatment (*Figure 3—figure supplement 1D*), suggesting that PP1 activity regulates the phosphorylation state of S229 in Oct4. These findings demonstrate that PP1 isoforms have an opposite activity with Aurkb by binding to Oct4 and by dephosphorylating S229 of Oct4 during the M/G1 transition in ESCs.

## PP1-mediated dephosphorylation of Oct4(S229) correlates with the resetting of pluripotency in the next G1 phase

Based on the findings that p-Oct4(S229) dissociates from condensed chromatin (*Figure 1*) and PP1 dephosphorylates p-Oct4(S229) during the M/G1 transition (*Figure 3*), We hypothesized that PP1-mediated dephosphorylation of Oct4(S229) is required for the resetting of Oct4 for pluripotency gene expression during the M/G1 transition. To examine this possibility, we arrested E14 ESCs in G2/M phase with nocodazole and released them into normal serum (*Figure 4A and B*). The high amounts of p-Oct4(S229) that accumulated in G2/M phase vanished quickly after G1 phase (after 1 hr release), reappeared at late S phase (after 6 hr release) and enriched at the next M phase (10 hr after release). In addition, to confirm weather Oct4 binding to target genes are regulated by phosphorylation dependent manner throughout the S-G2/M phase, we chased p-Oct4 level and binding of Oct4 to target genes (*Figure 4—figure supplement 1A and B*). Binding of Oct4 to target chromatin declined through S-G2/M phase (7–9 hr after release), and increased at the M/G1 transition (10 hr after release) in parallel with the level of p-Oct4(S229) accumulated. This result is consistent with recent report that Aurkb is active during S phase in ESCs (*Mallm and Rippe, 2015*).

Interestingly, the overall levels of Oct4 protein did not change significantly throughout the cell cycle. However, unlike Oct4, the levels of Nanog, which has a short half-life (*Ramakrishna et al., 2011*), rose in G1 phase, declined through S-G2/M phase, and reappeared at the start of the next G1 phase. This finding indicates that the synchronization of the phosphorylation state of Oct4(S229) with the cell cycle is linked to the resetting of Oct4 to its target genes.

To test that PP1 is required for resetting the transcription of Oct4 target genes, we administered 50 nM OKA to nocodazole-pretreated E14 ESCs for 4 hr and released them into normal serum. As expected, OKA retarded the dephosphorylation of p-Oct4(S229) and re-entry into the next G1 phase (*Figure 4C and D*).

To examine the binding of Oct4 to its targeting pluripotency genes on chromatin during the M/G1 transition, we performed the ChIP-qPCR assay. As a result, Oct4 bound weakly to target genes in G2/M phase, strengthening its association during entry into G1 phase. On treatment with OKA, Oct4 binding to target genes declined significantly during entry into G1 phase (*Figure 4E*). In

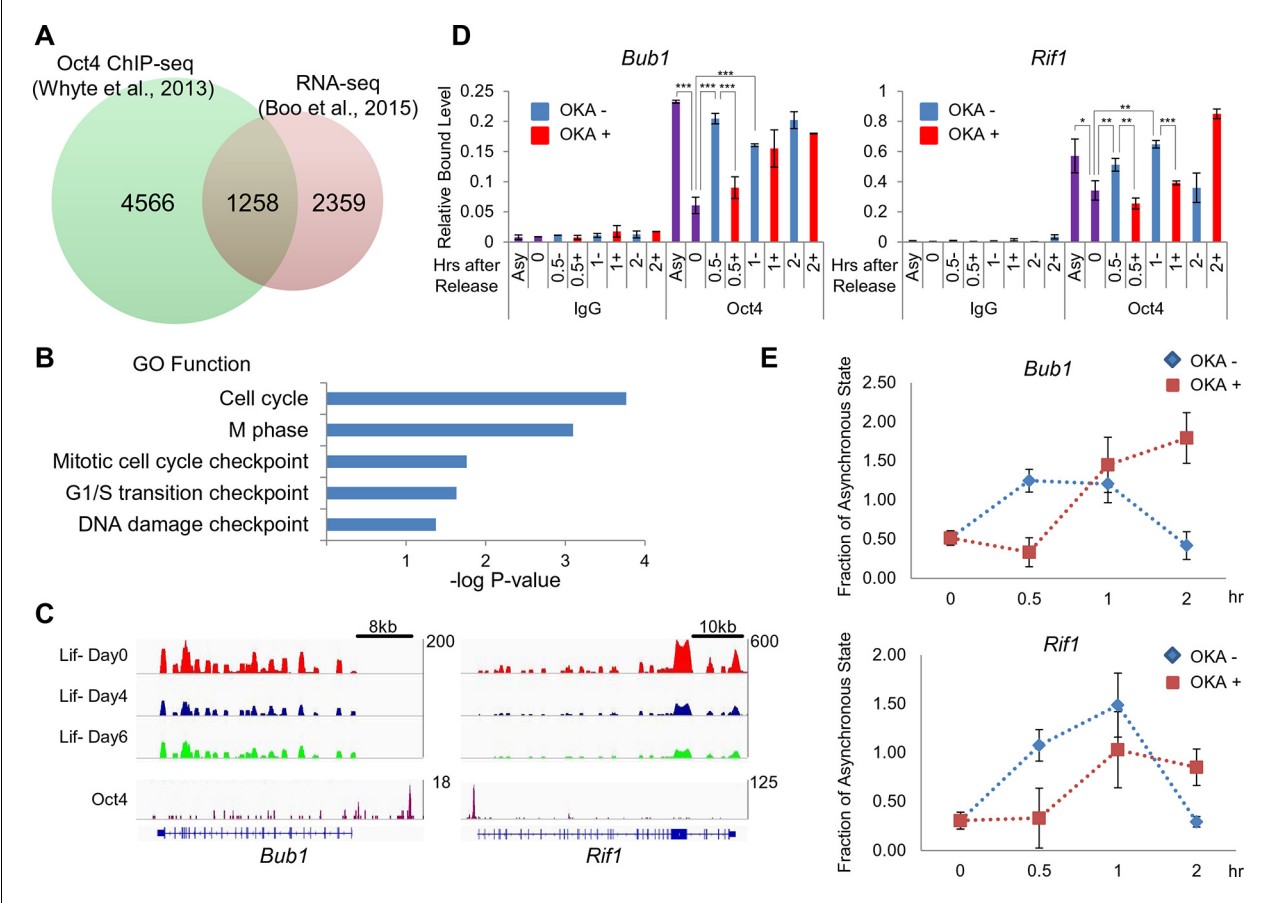

**Figure 5.** Oct4 regulates cell cycle related genes by direct targeting and resetting during the M/G1 transition. (A) A Venn diagram shows overlapped genes between proximal genes of Oct4 binding sites (green, n=5824; (*Whyte et al., 2013*)) and downregulated genes (fold changes≤0.75) in ZHBTc4 ESCs after Oct4 depletion by doxycycline treatment for 2 days (red, n=3617; [*Boo et al., 2015*]). (B) Gene ontology (GO) functional categories for putative Oct4 target genes. Cell cycle related GO functional categories are enriched. (C) RNA-seq reads of *Bub1* and *Rif1* of E14 ESCs during ESC differentiation upon LIF withdrawal (upper panel; [*Xiao et al., 2012*]) and ChIP-seq binding profiles of Oct4 at the *Bub1* and *Rif1* locus in undifferentiated E14 ESCs (lower panel; [*Whyte et al., 2013*]). (D) ChIP-qPCR analysis of E14 ESCs with anti-Oct4 in regions of *Bub1* and *Rif1* during the M/G1 phase transition with or without OKA treatment. IgG was used as a control. Values represent mean ± standard deviation (n≥3). t-test was used to calculate the statistical significance of differences in enrichment levels of Oct4 in ESCs during the M/G1 transition with or without OKA treatment. (*p<0.05, **p<0.01, ***p<0.001) (E) Nascent RNA levels of *Bub1* and *Rif1* in E14 ESCs were nalyzed by real-time qPCR during the M/G1 phase transition with or without OKA treatment. Levels of nascent RNA were divided by those in asynchronous state of E14 ESCs.

The following source data is available for figure 5:

**Source data 1.** Identification of putative Oct4 target genes.

**Source data 2.** Cell-cycle related genes in putative Oct4 target genes.

addition, to elucidate the cell cycle effect induced by OKA treatment to Oct4 binding, we performed ChIP-qPCR assay in Zhbtc4 ESCs stably expressing wild-type Oct4 (WT) and phosphor-defect mutant (S229A). When cells were released into G1 phase, OKA treatment significantly prevented the binding of Oct4(WT) to target genes. On the other hand, binding of Oct4(S229A) was relatively less affected even in treatment of OKA (*Figure 4—figure supplement 1C*). However, Oct4(S229A) mutant affects the O-GlcNAcylation of Oct4, which is critical for Oct4 activity, thereby Oct4(S229A) mutant fail to self-renew (*Jang et al., 2012*).

To analyze Oct4-depenent transcriptional resetting during the M/G1 transition, we measured nascent RNA levels (*Figure 4F*). When E14 ESCs were arrested in G2/M phase, nascent RNA levels of a subset of Oct4-targeting pluripotency genes declined significantly versus asynchronous ESCs.

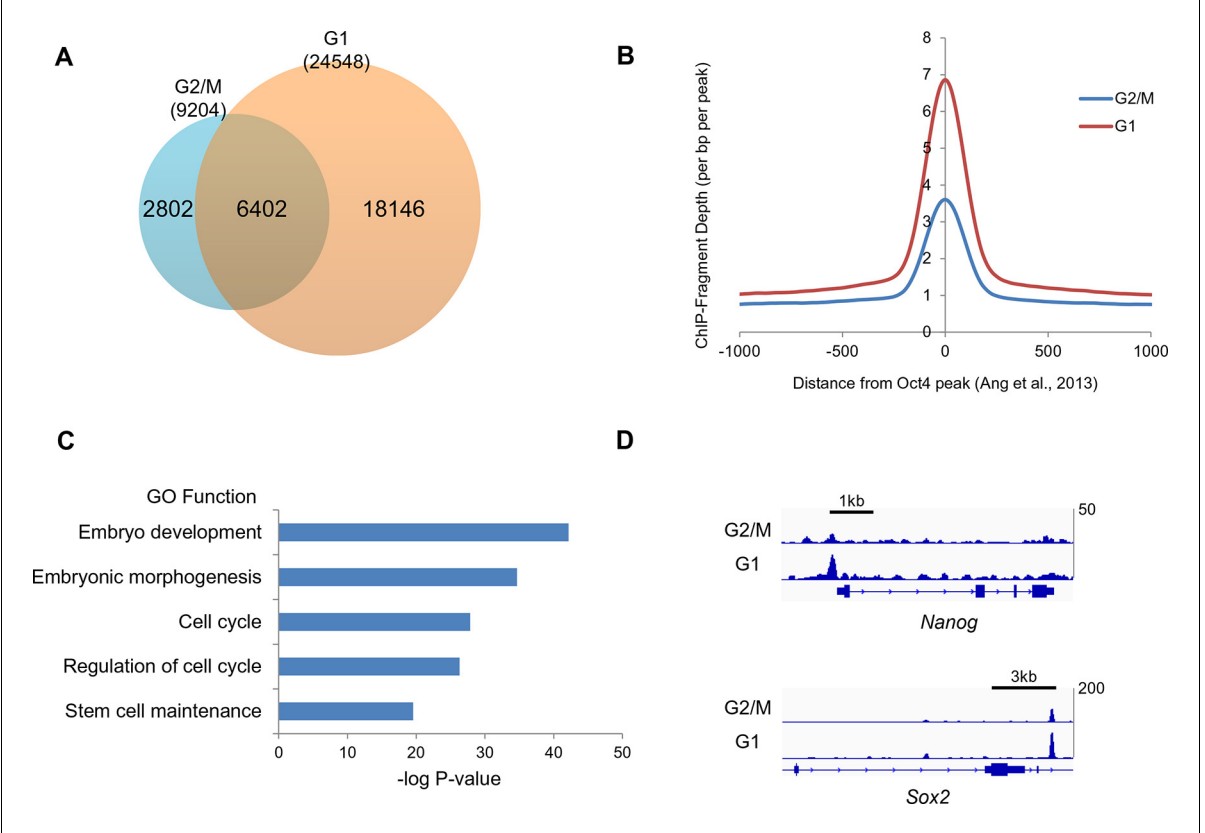

**Figure 6.** Oct4 ChIP-seq at G2/M and G1 phase of cell cycle. (**A**) Venn diagram of overlap between G2/M and G1 ChIP-seq peaks. (**B**) Mean ChIP-seq density of Oct4 around previously published Oct4-occupied regions (***Ang et al., 2011***) between G2/M and G1 phase. The level of Oct4 increased after release into M/G1 transition compared to G2/M phase. (**C**) Gene ontology (GO) functional categories for genes which are reset by Oct4. Pluripotency and cell cycle related GO functional categories are significantly enriched. (**D**) Integrated genomics viewer (IGV) screenshots for ChIP-seq data of Nanog and Sox2.

The following source data is available for figure 6:

**Source data 1.** Identification of G2/M or G1 specific Oct4 binding peaks.
**Source data 2.** Candidate genes reset by Oct4.

Nascent RNA levels of certain Oct4 target genes were upregulated when cells entered G1 phase (until 2 hr after release). Complementing the ChIP data, the nascent RNA levels of target genes were retarded after OKA treatment. Thus, we conclude that dephosphorylation by PP1 is critical for the transcriptional resetting of Oct4 to pluripotency genes during the M/G1 transition.

## Oct4 targets and resets cell cycle related genes in the next G1 phase

We next wondered whether Oct4 resets a subset of cell cycle related genes during the M/G1 transition. To address this, we first narrowed down the putative 1258 Oct4 target genes by crossover between 5824 genes co-occupied by OSN and 4617 genes decreased by Oct4 depletion in ZHBTc4 ESCs (***Figure 5A*** and ***Figure 5—source data 1***) using publically-available ChIP-seq and RNA-seq data (***Boo et al., 2015***; ***Whyte et al., 2013***). By gene ontology analysis of the putative Oct4 target genes using DAVID (http://david.abcc.ncifcrf.gov), we identified some Oct4 target genes associated with various cell cycle related functional categories (***Figure 5B*** and ***Figure 5—source data 2***).

Intriguingly, we found that Oct4 governs the cell cycle genes related to S/G2/M phase. Thus, among these genes related to S/G2/M phase, we focused on Bub1 and Rif1 because loss of Bub1 and Rif1 are known to induce differentiation in the ESC-based knockdown experiment (***Dan et al., 2014***; ***Lee et al., 2012***) and expression levels of both genes decrease upon ESC differentiation

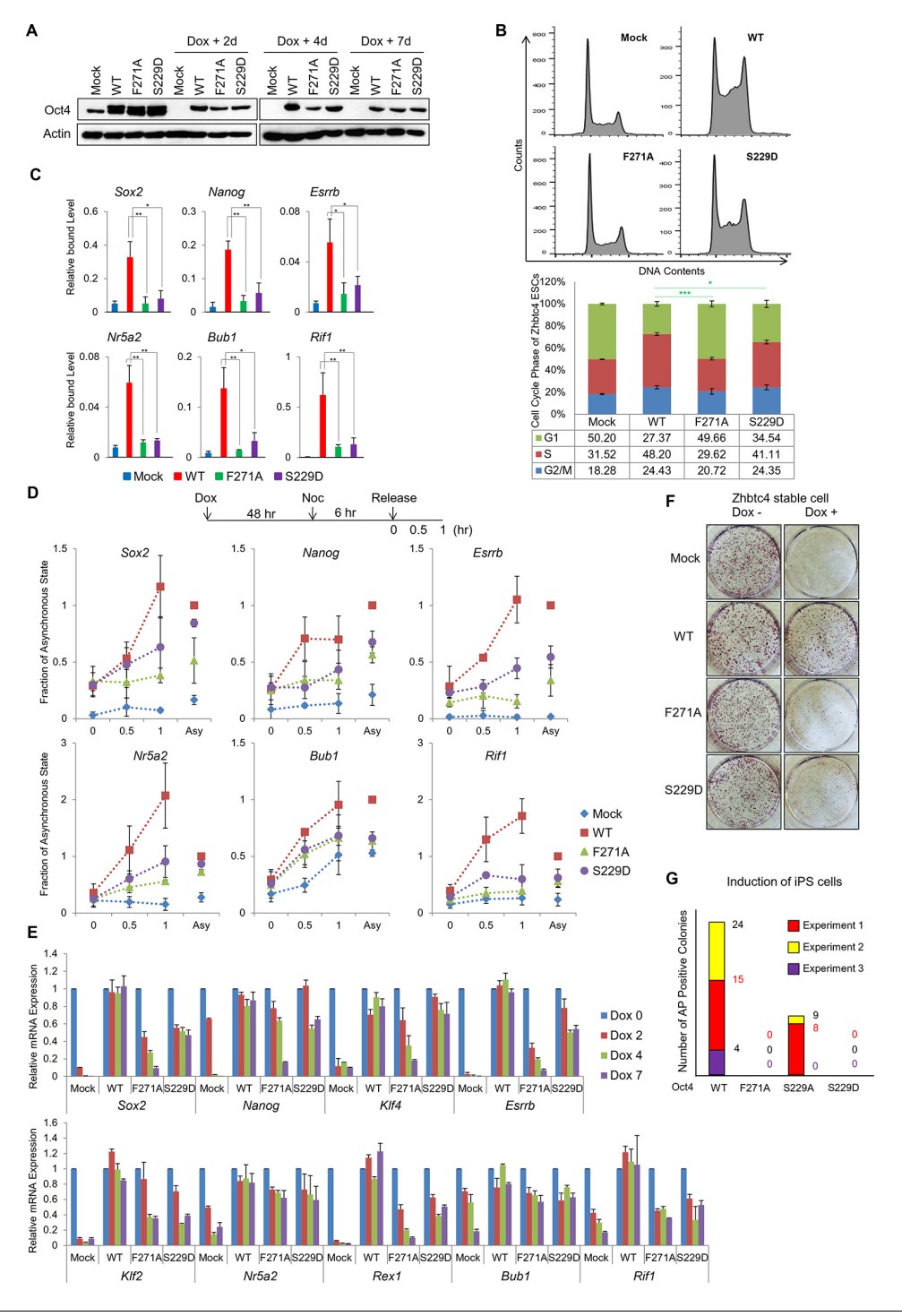

**Figure 7.** Oct4 mutants—Oct4(S229D) and Oct4(F271A)—effect the loss of pluripotency and alter the cell cycle by impeding gene expression. (**A**) Flag Oct4 WT and mutants were stably incorporated into the genome of ZHBTc4 ESCs. These cells were treated with doxycycline for the indicated days. Stable expression of exogenous Oct4 was confirmed by western blot. (**B**) After 2 days of doxycycline treatment in ZHBTc4 ESCs (Mock, wild-type Oct4(WT)-, Oct4(F271A)-, and Oct4(S229D)- backup cells), DNA content ($1\times10^4$ cells/sample) were analyzed in each ESC by FACS. Values represent mean ± standard deviation (n≥3). t-test was used to calculate the statistical significance of differences in G1 phase of ZHBTc4-Oct4(WT) versus -Oct4(S229D) and -Oct4(F271A). (*p<0.05, ***p<0.001) (**C**) Anti-Oct4 ChIP-qPCR of ZHBTc4 ESCs that express exogenous Oct4 at 2 days after doxycycline treatment. F271A and S229D mutants showed decreased binding to target genes compared with Oct4 WT. Oct4-depleted ZHBTc4

*Figure 7 continued on next page*

*Figure 7 continued*

cells (Mock) were used as a control. Values represent mean ± standard deviation (n≥3). t-test was used to calculate the statistical significance of differences in enrichment levels of Oct4 in ZHBTc4-Oct4(WT) versus Oct4(S229D) and Oct4(F271A) ESCs. (*p<0.05, **p<0.01) (**D**) Levels of nascent RNA were measured by qRT-PCR and each nascent RNA levels were normalized by the levels in Oct4(WT) backup ZHBTc4 ESCs in the asynchronous state after doxycycline treatment for 2 days. The experimental scheme is shown (upper panel). (**E**) Relative expression of genes targeted by Oct4 related to pluripotency and cell cycle in ZHBTc4. mRNA levels of the indicated genes decreased significantly in Mock, Oct4(F271A), and Oct4(S229D) backup cells after doxycycline treatment but not in Oct4(WT) backup cells. (**F**) Indicated ZHBTc4 ESCs were stained for alkaline phosphatase (AP) activity after 7 days of doxycycline treatment. (**G**) Reprogramming of MEFs into iPS cells driven by Oct4, Sox2, and Klf4. Oct4 wild-type was replaced by F271A, S229A, and S229D mutants. Reprogrammed cells were identified by AP staining and counted. Results from 3 independent experiments are presented.

The following figure supplement is available for figure 7:

**Figure supplement 1.** Both S229D and F271A mutants of Oct4 decrease Oct4 activity.

---

(*Figure 5C* upper) in previously published RNA-seq study (*Xiao et al., 2012*). Furthermore, both Oct4 enrichment score and expression level of *Bub1* and *Rif1* were one of the top10 genes among putative Oct4 targeting cell cycle related genes (lower, *Figure 5C* and *Figure 5—source data 2*).

To investigate whether Bub1 and Rif1 are reset by PP1-mediated dephosphorylation of Oct4 during the M/G1 transition, we first investigated the resetting of Oct4 to target genes during the M/G1 transition by ChIP-qPCR (*Figure 5D*). Oct4 binding to both genes on chromatin was weakly sustained in G2/M phase and rapidly increased during the M/G1 transition (0.5 hr after release into normal serum), and thereafter Oct4 binding is either saturated or declined, while induced Oct4 binding to both genes during the M/G1 transition were relatively retarded when cells were released into normal serum with treatment of OKA (*Figure 5D*). Likewise, nascent RNA levels of *Bub1* and *Rif1* were downregulated when cells were arrested in G2/M phase and upregulated during the M/G1 transition, but rapid increase of nascent RNA levels of both genes during entry into G1 phase was retarded by OKA treatment (*Figure 5E*). Considered together, we concluded that Oct4 directly controls cell cycle related genes by its resetting to cell cycle related genes during the M/G1 transition.

## Oct4 ChIP-seq reveals that resetting regions enriched in pluripotency and cell cycle related genes

To identify which regions are reset by Oct4, we mapped the genomewide occupancy of Oct4 at G2/M and G1 phase in E14 ESCs by ChIP-seq. We found high-confidence peaks (with p value<$10^{-5}$) at G2/M (9204) and G1 (24548) phase (*Figure 6A* and *Figure 6—source data 1*). Expectedly, Oct4 bound much more regions of genome at G1 phase rather than G2/M phase (*Figure 6A*). Furthermore, mean peak density of Oct4 bound regions was higher in G1 phase than G2/M phase (*Figure 6B*) implicating that Oct4 genome-widely resets the target genes at G1 phase. Next to identify reset region by Oct4, we discovered 9092 of genomic regions enriched by Oct4 more than twofold increase in G1 phase compared to G2/M phase (*Figure 6—source data 2*). We categorized these as the Oct4 resetting peaks, which are significantly enriched in not only pluripotency but also cell cycle categories (*Figure 6C*). For example, we showed that Oct4 more strongly binds to the regions of both Nanog and Sox2 at G1 phase rather than G2/M phase, supporting that Oct4 resets its target genes at G1 phase (*Figure 6D*).

## Oct4 mutants—Oct4(S229D) and Oct4(F271A)—effect the loss of pluripotency and alter the cell cycle by impeding gene expression

To determine the significance of recycling Oct4 through Aurkb/PP1 in ESC pluripotency, we generated ZHBTc4 ESCs that stably expressing wild-type Oct4(WT), a phosphor-mimic Oct4(S229D) mutant, and a PP1-binding-defective Oct4(F271A) mutant. We confirmed that ectopic wild-type Oct4 and Oct4 mutants were expressed when endogenous Oct4 was removed by doxycycline (dox) treatment (*Figure 7A*).

Under the same conditions (dox for 2 days), we analyzed the cell cycle patterns in these ESCs (*Figure 7B*). Oct4-depleted ZHBTc4 ESCs (Mock) harbored significantly more cells in G1 phase and

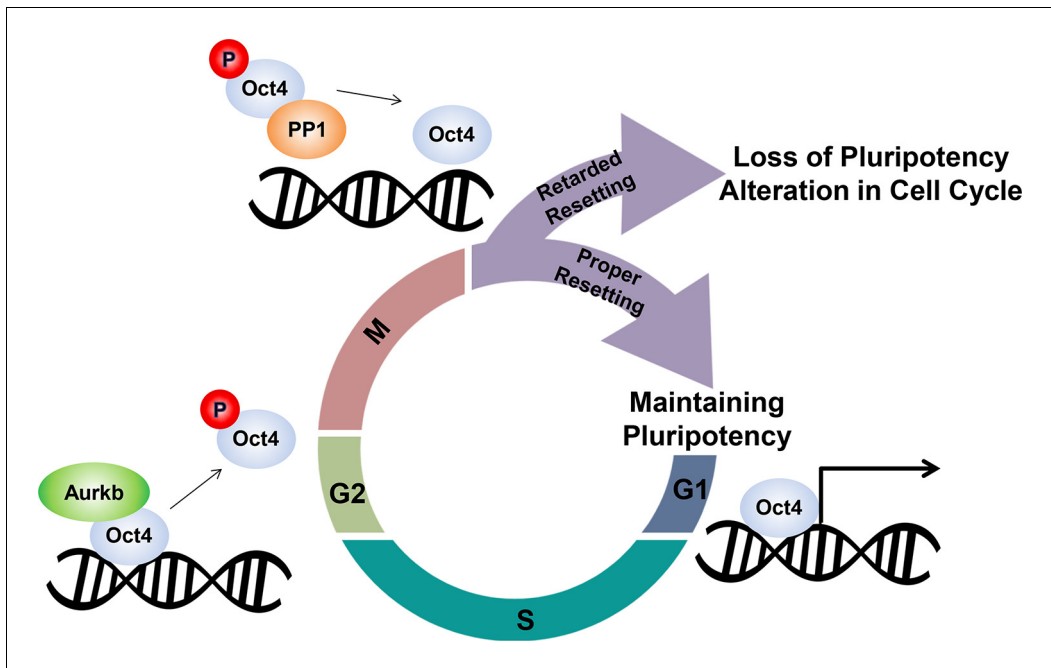

**Figure 8.** Schematic. A model describing the dissociation and resetting of Oct4 on chromatin by Aurkb/PP1 during the cell cycle. Aurkb phosphorylates Oct4(S229), leading to dissociation of Oct4 from chromatin during G2/ M phase. On mitotic exit, PP1 binds to Oct4 and dephosphorylates Oct4(S229), which resets Oct4-driven transcription to maintain pluripotency and cell cycle progression.

fewer S-phase populations, whereas Oct4(WT)-backup cells had a typical ESC cell cycle profile. Notably, the cell cycle profile of Oct4(F271A)-backup cells resembled that of ZHBTc4 ESCs (Mock). Oct4(S229D)-backup cells contained larger G1-phase populations than Oct4(WT)-backup cells but fewer than Oct4(F271A)-backup cells, indicating that the PP1-binding motif in Oct4 is essential for maintaining the pluripotency and cell cycle progression of ESCs. The binding of Oct4 mutants to target genes—including pluripotency-related and cell cycle genes—decreased significantly versus Oct4 (WT) after dox treatment for 2 days (*Figure 7C*).

To determine the reset patterns of Oct4 mutants during the M/G1 transition, we analyzed nascent RNA levels of a subset of Oct4 target genes in ZHBTc4 ESCs harboring Oct4(WT) and Oct4 mutants that were pretreated with dox for 2 days, given nocodazole (6 hr), and released into normal serum (*Figure 7D*).Nascent RNA transcripts of certain Oct4 target genes in Oct4(WT)-backup cells were upregulated during the M/G1 transition, whereas those in Oct4-mutated cells did not increase versus ZHBTc4 Oct4(WT), indicating that Oct4 mutations (S229D, F271A) impede the prompt resetting of Oct4 during the entry into the next G1 phase.

To determine the long-term effects of Oct4 mutations on ESC pluripotency and cell cycle progression, we cultured ZHBTc4 ESCs containing Mock, Oct4(WT), Oct4(S229D), and Oct4(F271A) for 7 days with dox. Ectopic Oct4(WT) and Oct4 mutants (S229D, F271A) were continuously expressed until dox treatment for 7 days (*Figure 7A*). Total mRNA expression of a subset of Oct4 target genes in Oct4(WT) cells was continuous, whereas that in Oct4-depleted and Oct4 mutant-backup cells was significantly downregulated (*Figure 7E*). Consistent with this finding, alkaline phosphatase (AP) positive colonies in Oct4-depleted and Oct4 mutant-backup cells was much lower than in Oct4(WT)-backup cells (*Figure 7F*).

To determine the effects of mutation on Oct4 transcriptional activity, we infected retroviral wild-type and mutant Oct4 into NIH-3T3 cells in which a 10X Oct4 response element (RE)-driven luciferase reporter was stably incorporated and measured luciferase activity. Oct4(F271A) and Oct4 (S229D) showed little luciferase activity (*Figure 7—figure supplement 1A*). Oct4(S229A) also had weaker activity than Oct4(WT). Consistent with these data, during Oct4 depletion in ZHBTc4 ESCs, retroviral infection with Oct4(F271A) and Oct4(S229D) failed to rescue the maintenance of ZHBTc4

ESC self-renewal compared with Oct4(WT) infected cells (*Figure 7—figure supplement 1B and C*). In addition, these Oct4 mutants impeded somatic cell reprogramming (*Figure 7G*).

Based on our findings, we propose that the recycling of Oct4 by Aurkb/PP1 over time and by location is pivotal for the transcriptional resetting of Oct4 during entry in to the subsequent G1 phase, ultimately maintaining ESC pluripotency and cell cycle progression (*Figure 8*).

## Discussion

In this study, we have demonstrated that Oct4, a master pluripotency transcription factor, is spatio-temporally regulated by the Aurkb-PP1 axis during the cell cycle. In G2/M phase, Aurkb phosphorylates Oct4 extensively at serine 229, leading to its dissociation from chromatin (*Figure 1* and *Figure 2*). The detachment of Oct4 from chromatin is consistent with findings from previous reports. Most transcriptional machinery proteins, such as RNA pol II (*Gottesfeld and Forbes, 1997*; *Parsons and Spencer, 1997*), and many transcription factors dissociate from condensed chromatin at the onset of mitosis, and many studies have implied that mitotic dissociation of transcription factors occurs through phosphorylation (*Delcuve et al., 2008*; *Dephoure et al., 2008*).

Aurka regulates ESC pluripotency through phosphorylation-mediated inhibition of p53 (*Lee et al., 2012*) but is not involved in the phosphorylation-mediated recycling of Oct4 (*Figure 2* and *Figure 2—figure supplement 1*). Thus, both Aurk isoforms—Aurka and Aurkb—appear to be related to ESC pluripotency, but in a different molecular mechanism.

We also found that PP1 mediates the resetting of Oct4 during the M/G1 transition. PP1 governs the resetting of cell cycle machinery (*Ceulemans and Bollen, 2004*), binding specific sequences—the RVxF motif—and dephosphorylating interactors (*Hendrickx et al., 2009*). We identified an RVWF motif in the C-terminal POU-h domain (*Figure 3A*) and an Aurkb phosphorylation site (S229) in the N-terminal POU-h domain of Oct4, which come into close proximity in the 3-dimensional structure (*Figure 3B*). This arrangement convinces us that PP1 binds Oct4 through the RVxF motif and dephosphorylates Oct4 in vitro (*Figure 3E and F*).

In addition, we revealed that PP1 inhibition by OKA treatment delayed the Oct4 dephosphorylation at S229 residue and consequently resetting to the target genes in E14 ESCs during M/G1 transition (*Figure 4C and E*). We further found that OKA treatment relatively less affected the binding of phosphor-defect Oct4(S229A) to target genes, significantly impeding the resetting of Oct4(WT) at M/G1 transition (*Figure 4—figure supplement 1C*), implying that dephosphorylation of S229 residue by PP1 is paramount for the resetting of Oct4 to target genes. Nonetheless, we cannot perfectly rule out the possibility that OKA treatment may affect the function of broad spectra of cell cycle regulators during M/G1 transition.

We confirmed that Aurkb-phosphor-mimetic and PP1-binding-defective mutations lead to a loss of pluripotency, even in the presence of LIF, and that introduction of these mutants into somatic cells lowers the reprogramming efficiency (*Figure 7G*), supporting that Aurkb/PP1 is a critical pair of regulators in resetting Oct4 on chromatin during the cell cycle.

Mitotic phosphorylation of a transcription factor affects its transcriptional activity and other properties. For example, Oct1, a member of the POU domain transcription factors, is phosphorylated during mitosis and localizes to the spindle matrix, forming a complex with lamin B1 at the midbody (*Kang et al., 2011*). Mitotic phosphorylation of Sp1 protects itself from ubiquitin-dependent degradation (*Chuang et al., 2008*). Notably, we found that Oct4 protein levels were unchanged, even though nascent *Oct4* RNA levels fluctuated during the cell cycle (*Figure 4A and F*), implying that phosphorylated Oct4 is probably protected from degradation during the cell cycle. We are interested in determining the mechanism of how phosphorylated Oct4 escapes protein degradation.

The molecular association between pluripotency and the cell cycle in ESCs has garnered attention with regard to identifying the mechanism by which the cell fate of ESCs is determined. In particular, the protraction of G1 in naive ESCs by knockdown of cyclin E1 causes spontaneous differentiation; in contrast, promoting G1/S transition through overexpression of cyclin E1 enhances self-renewal (*Coronado et al., 2013*). G1 phase regulators, cyclin D proteins control the differentiation of hESCs into various lineages via the TGF-β-Smad2/3 pathway (*Pauklin and Vallier, 2013*). Sox2 was identified as a cell cycle regulator that suppresses p21 and p27 and induces cyclin D3 expression (*Her-reros-Villanueva et al., 2013*). Oct4 downregulation lengthens G1 phase and upregulates p21 in ESCs (*Lee et al., 2012*). A nontranscriptional function of Oct4 in mitotic entry has recently been

reported (*Zhao et al., 2014*). A recent study performed functional screening of human embryonic stem cells under various differentiation conditions and identified that genes involved in S and G2 phases are gatekeepers of differentiation (*Gonzales et al., 2015*). Furthermore, S and G2 phases of cell cycle possess an intrinsic propensity for maintaining pluripotency.

In this study, in addition to observing that Aurkb/PP1 control the dynamics of Oct4 during the cell cycle, we found that Oct4 governs the cell cycle of ESCs by directly targeting genes that are related to cell cycle regulation and pluripotency. In particular, 2 cell cycle genes—*Bub1* and *Rif1*— are reset by Oct4 during the cell cycle (*Figure 5E* and *Figure 7D*). Considering that Bub1 and Rif1 are important cell cycle regulators that control the checkpoints for mitosis and replication (*Bolanos-Garcia and Blundell, 2011*; *Yamazaki et al., 2013*) and are crucial for maintaining pluripotency (*Dan et al., 2014*; *Lee et al., 2012*), the regulation of S-G2-M phase by Oct4 might also be critical for ESC pluripotency.

Our study might provide insights into why ESCs reset Oct4 during the cell cycle. Like, this mechanism might bifurcate the fate of ESCs: tight resetting of Oct4 on chromatin during the cell cycle strengthens the pluripotency of ESCs at the ground state, but on differentiation, loose resetting of Oct4 at the next M/G1 transition leads to lineage differentiation.

During reprogramming by nuclear transfer, mitotic chromosomal condensation is required to reset the origins of replication of differentiated donor cells in embryonic DNA replication (*Lemaitre et al., 2005*), transfer of a mitotic genome into a zygote in mitosis-enhanced reprogramming (*Egli et al., 2007*), and mitotic chromatin induces core pluripotency factors more rapidly than interphase nuclei (*Halley-Stott et al., 2014*), suggesting that a genome can be exchanged during mitosis which is an open window that allows transcription factors to occupy target genes on mitotic exit and thus enabling postmitotic cell fate changes to be induced.

We have provided evidence that cell cycle machinery cooperates with pluripotency transcriptional programs. The resetting of Oct4 occurs rapidly during the exit from mitosis and that delayed resetting alters the cell cycle and effects the loss of pluripotency—ie, prompt resetting of Oct4 prevents postmitotic cell fate changes in ESCs. Based on our results, we suggest that the potential of ESCs to differentiate might be derived from the small window of the M/G1 transition by which the resetting of Oct4 is the central mechanism to determine the maintenance of ESC pluripotency or lineage commitments.

## Materials and methods

### Cell culture

E14 and ZHBTc4 ESCs were cultured as described (*Jang et al., 2012*) in 0.1% gelatin (Sigma-Aldrich, St. Louis, Missouri ) coated plates. ZHBTc4 ESCs were kindly provided by Hitochi Niwa (RIKEN, Japan). The mouse ESC medium was composed of DMEM (Hyclone, Logan, Utah) and 15% (v/v) fetal bovine serum (FBS; Gibco, Grand Island, New York), supplemented with 2 mM L-glutamine, 55 µM β-mercaptoethanol, 1% (v/v) nonessential amino acids, 100 U/ml penicillin, 100 µg ml$^{-1}$ streptomycin (all from Gibco), and 1000 U/ml ESGRO (Millipore, Germany). Aurora kinase inhibitors AT9283, Hesperadin and MLN8237 were purchased from Selleckchem (Houston, Texas) and okadaic acid from Sigma.

### Mitotic arrest-release and cell cycle analysis

For G2/M phase synchronization, E14 cells were treated with 200 ng/ml nocodazole (Calbiochem, Germany ) for 10 hr and ZHBTc4 cells were treated with same concentrations of nocodazole for 6 hr. In order to release, synchronized cells were washed three times with PBS, and incubated for the indicated time in fresh culture media.

For cell cycle analysis, the collected cells at the indicated time were immediately fixed in 70% ethanol and stained with propidium iodide (PI; Sigma, P4170) for 1 hr at room temperature in the dark. Cell cycles were analyzed using FACSCalibur flow cytometer and LSRII (SORP) (Becton Dickinson, Franklin Lakes, New Jersey). Analysis of cell cycle data was performed with FlowJo (Tree Star Inc., Ashland, Oregon).

## Immunofluorescence and confocal microscopy

Cells grown on coverslips were fixed in 4% (w/v) paraformaldehyde and permeabilized in 0.5% (w/v) Triton X-100 in PBS for 30 min at room temperature (RT). After permeabilization, the cells were blocked with 3% (w/v) BSA for 30 min. Subsequently, they were incubated in primary antibody for 1 hr at RT. Antibody dilutions were 1:500 for anti-Oct4 (Santa Cruz, Dallas, Texas, sc-5279), 1:200 for anti-p-Oct4(S229), 1:200 for pH3S10 (Millipore, 05–598). Secondary antibodies used in immunostaining were Alexa Fluor 488, 568 (Invitrogen, Carlsbad, California).

Confocal micro-images were obtained by a confocal laser scanning microscope (Carl Zeiss, Germany, LSM 510 META).

## DNA constructs

The plasmid pGAE-mKO2:Cdt1 and pGAE-mAG:Geminin were generously provided by Savatier, P. (INSERM U846, France). For stable expression in ESCs, fragments containing mKO2:Cdt1 and hmAG1:Geminin coding sequences, respectively, were generated by PCR amplification and were sub-cloned between the SalI and AgeI sites into pCAG-IP vector to generate pCAG-mKO2:Cdt1-IP and pCAG-mAG:Geminin-IP. PP1$\alpha$, $\beta$ and $\gamma$ were amplified by PCR from cDNA of E14 ESCs. The PCR products were digested with XhoI and AgeI then subcloned into pCAG-Flag-IP vector. All Oct4 mutants were generated by site-directed mutagenesis (Intron, Korea).

## Generation of stably expressing Flag-Oct4 ZHBTc4 ESCs

For long-term transgene expression in ZHBTc4 ESCs, Flag-tagged Oct4 was cloned into pCAG-Flag-IP, which was generated by inserting a Flag tag into pCAG-IP, kindly provided by Hitoshi Niwa (RIKEN, Japan). To generate ESCs stably expressing Flag-Oct4, pCAG-Flag-Oct4-IP was transfected into ESCs using Lipofectamine (Invitrogen). After 48 hr of transfection, selection with 2µg/ml of puromycin was performed to determine stable integration. Puromycin resistant cells were expanded and analyzed for Oct4 by Western blot.

## Reporter gene assay

The reporter gene assay was done as described (*Jang et al., 2012*). Briefly, Ten copies of Oct4-responsive element(10X Oct4 RE)-driven luciferase reporter gene was incorporated into the genome of NIH 3T3 cells by retroviral infection. To stably incorporate reporter gene into genomic DNA, cells were selected with puromycin for at least 2 weeks. These stable cells were infected with retroviruses expressing Oct4. Luciferase activity was measured 4 days after infection of Oct4.

## In vitro kinase assay

0.2 µg of purified GST-Oct4 proteins were used in cold in vitro kinase assays with purified recombinant kinases. All kinases were purchased from Proqinase (Germany). Kinase reactions were performed in kinase buffer (60mM HEPES-NaOH pH 7.5, 3 mM MgCl2, 3 mM MnCl2, 3 µM Na-orthovanadate, 1.2 mM DTT, 0.5mM ATP) for 30 min at 30℃. Then reactions were stopped by the addition of 5X SDS-PAGE loading buffer and assessed by Westernblot.

For radioactive in vitro kinase assay GST-Oct4 was incubated with Aurkb in kinase buffer (60 mM HEPES-NaOH pH 7.5, 3 mM MgCl2, 3mM MnCl2, 3 µM Na-orthovanadate, 1.2 mM DTT, 0.25 mM ATP) with 0.1 µM $\gamma$-$^{32}$P-ATP (NEG002A250UC, purchased from PerkinElmer, Waltham, Massachusetts) for 30 min at 30℃. Reactions were then stopped by the addition of 5X SDS-PAGE loading buffer and loaded for separation on 8% SDS-PAGE gel. After staining with Coomassie Blue, the gels were dried and exposed to films.

## In vitro phosphatase assay

In vitro phosphatase assay was performed as described (*Kassardjian et al., 2012*). Briefly, phosphorylated GST-Oct4 was pulled down with Glutathione Sepharose 4B (Sigma) for 4 hr at 4℃ and suspended in phosphatase buffer. GST-Oct4 beads phosphorylated by Aurkb were incubated with 1 µg of purified His-tagged PP1 for 1 hr at 30℃, with mild shaking.

## Immunoprecipitation and Western blot

Immunoprecipitation and Western blot were performed as described (*Jang et al., 2012*). Anti-p-Oct4(S229) antibody was made by GenScript (Piscataway, New Jersey). Anti-Oct4 (sc-5279), anti-PP1α (sc-6104) and anti-PP1γ (sc-6108) were acquired from Santa Cruz Biotechnology; anti-Nanog (ab14959) and anti-PP1β (ab53315) were purchased from Abcam (UK); anti-Aurka (610938) and anti-Aurkb (611082) were acquired from BD Transduction Laboratories (San Jose, California); anti-Esrrb (H6707) was obtained from R&D System (Minneapolis, Minnesota); and anti-phospho-Histone H3 (Ser10) (05–598) was purchased from Millipore.

## Nascent RNA analysis

To prepare nascent RNA, Click-iT Nascent RNA Capture Kit (Life Technologies, Carlsbad, California) was used. First, ESCs were incubated with 100 μM of 5-ethynyl uridine (EU) for 15 min. After incubation, EU-labeled RNA was isolated and converted into biotinylated RNA by Click reaction. The biotinylated RNA was pulled down with streptavidin magnetic beads. cDNA was synthesized with RNA bound to the beads as a template and analyzed by qRT-PCR. The primers that span an intron–exon boundary were used for specifically detecting of nascent RNA. The primer sequences used in the analysis are given in *Supplementary file 1*.

## Lentiviral shRNA-mediated knockdown

Lentiviruses were produced using PLKO-puro constructs that express shRNAs; Aurka#2 (TRCN0000025140), Aurka#5(TRCN00000251430), Aurkb#2(TRCN0000374361), Aurkb#3 (TRCN0000321718), Pak1#4(TRCN0000025258), Pak2#1(TRCN0000025209), Pak2#5 (TRCN0000413412), Pkca#2(TRCN0000022875) and Pkca#5(TRCN00002187830) were purchased from Sigma. 293FT cells were cotransfected with 0.5 μg each of pMD2.G, pMDLg/pRRE, pRSV-rev, and 1 μg of pLKO-shRNA using Lipofectamine (Invitrogen) in a 6-well plate. 48 hr after transfection, virus-containing medium was collected and passed through 0.45-μm filters. Polybrene (10 μg/ml) was added to target cells immediately prior to infection, and infection was performed for 5 hr. Target cells were selected with puromycin (2 μg/ml) 48 hr after infection.

## Real-Time qPCR, chromatin immunoprecipitation (ChIP) assay

Preparation of RNAs, reverse transcription PCR, real-time qPCR, and chromatin immunoprecipitation (ChIP) assay were done as described (*Jang et al., 2012*). The primer sequences used are given in *Supplementary file 1*.

## ChIP-seqeuncing

We ChIPed with Oct4 antibody at G2/M (nocodazole-treated cells) and G1 phase (release of nocodazole-treated cells) in E14 ESCs. ChIPed DNAs were sequenced by LAS (Korea, http://www.las-cience.co.kr). For the ChIP-seq analysis, reads were mapped to the mouse genome (NCBI build 37/mm9). The detailed analysis was done as described (*Kim et al., 2015*).

## Self renewal assay

Self-renewal assay (colony-forming assay) was performed as described (*Chambers et al., 2007*). ESCs were trypsinized to a single cell and re-plated 500 cells in a well of 6-well plates. After incubation for 7 days with/without doxycycline, the plates were stained for alkaline phosphatase and counted.

## Reprogramming

Reprogramming was performed as described (*Jang et al., 2012*). Briefly, equal amounts of retrovirus encoding Oct4, Sox2, and Klf4 were applied to MEFs in 10% FBS DMEM media containing 8 ng/ml polybrene. After 24 hr, fresh mESC culture media were added, and the culture was then maintained for up to 21 days. Reprogrammed cells were identified by alkaline phosphatase (AP) staining and scored.

Nano-LC-ESI-MS/MS Analysis of Phosphorylation Sites in Oct4.

Nano-LC-ESI-MS/MS analysis was previously performed as described (*Jang et al., 2012*).

## Acknowledgements
We thank Pierre Savatier (INSERM, France) for the FUCCI vectors and Dae-Sik Lim (KAIST, Korea) for Aurora kinase constructs. This work was supported by a National Research Foundation of Korea grant, funded by the Korean government (MSIP) (No. 2012R1A3A2048767 to H-DY) and by the Education and Research Encouragement Fund of Seoul National University Hospital. Jihoon Shin was supported by Health Fellowship Foundation.

## Additional information

### Funding

| Funder | Grant reference number | Author |
|---|---|---|
| National Research Foundation of Korea | No. 2012R1A3A2048767 | Hong-Duk Youn |

The funders had no role in study design, data collection and interpretation, or the decision to submit the work for publication.

### Author contributions
JS, Performed and the microscopic analysis, biochemical experiments and ChIP-seq exeriments, Conception and design, Acquisition of data, Analysis and interpretation of data, Drafting or revising the article; TWK, Wrote the manuscript; Performed the biochemical experiments, Conception and design, Acquisition of data, Analysis and interpretation of data; HK, Performed the biochemical experiments; Performed the ChIP-qPCR, Acquisition of data, Analysis and interpretation of data; HJK, Performed and the microscopic analysis; Performed the biochemical experiments, Acquisition of data, Analysis and interpretation of data; MYS, SL, H-TL, Performed the nascent RNA PCR, Acquisition of data; SK, Performed the bioinformatics analysis ChIP-seq, Analysis and interpretation of data; S-EL, HJ, Performed the ChIP-qPCR, Acquisition of data; J-HL, Performed the biochemical experiments, Acquisition of data; E-JC, Interpretated the nascent RNA transcription, Analysis and interpretation of data; H-DY, Supervised all the experiments and approved the final version, Conception and design, Drafting or revising the article

### Author ORCIDs
Hyonchol Jang, http://orcid.org/0000-0003-1436-457X
Hong-Duk Youn, http://orcid.org/0000-0001-9741-8566

## Additional files

### Supplementary files
• Supplementary file 1. The lists of primers for nascent RNA, real-time qPCR, and ChIP-qPCR in this study.

### Major datasets
The following dataset was generated:

| Author(s) | Year | Dataset title | Dataset URL | Database, license, and accessibility information |
|---|---|---|---|---|
| Jihoon Shin, Hong-Duk Youn | 2016 | | https://www.ncbi.nlm.nih.gov/geo/query/acc.cgi?acc=GSE78073 | The ChIP-seq datasets have been deposited into the NCBI Gene Expression Omnibus database under accession numbers GSE78073 |

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
