## [Decision Letter]

Thank you for submitting your work entitled "Aurkb/PP1-Mediated Resetting of Oct4 During the Cell Cycle Determines the Identity of Embryonic Stem Cells" for peer review at *eLife*. Your submission has been favorably evaluated by Fiona Watt (Senior editor), a Reviewing editor, and three reviewers.

The reviewers have discussed the reviews with one another and the Reviewing editor has drafted this decision to help you prepare a revised submission.

All reviewers found the work to be of considerable interest but wanted additional work to reinforce some of the most critical conclusions. Given that virtually nothing is known about how post-translational modifications regulate Oct4, this paper provides some provocative evidence that phosphorylation by Aurora B Kinase influences the association of Oct4 with chromatin. Addressing the critical points enumerated below will strengthen the paper:

1) Please determine when in the cell cycle Oct4 is bound to DNA and perform CHIP-seq during distinct phases of the cell cycle (e.g., G2M vs G1) to define which regions are bound in a cell-cycle dependent manner.

2) Please provide evidence for the robustness of the pOct4 Ab for CHIP.

3) Consider the criticism from Reviewer #2 regarding the lack of specificity of the OKA experiments.

4) Provide better characterization of the nocodozole effects as stressed by Reviewer #3.

*Reviewer #1:*

The manuscript describes data showing that Oct-4 is phosphorylated by aurora kinase B in the G2/M phase of the cell cycle of mouse embryonic stem cells (mESCs). This mechanism blocks Oct-4 to bind chromatin. PP1 then dephosphorylate Oct-4 in M/G1 restoring its transcriptional activity. Inhibition of this mechanism inhibits pluripotency and decrease cellular reprogramming efficiency. The authors propose that this mechanism could reset the pluripotent state of mouse ESCs at each cell cycle thereby allowing for a short period of time the cells to differentiate.

These data are interesting and novel. They offer new insights concerning the mechanisms by which cell cycle could control cell fate choice in stem cells. Nonetheless few aspects could be clarified.

The precise timing by which Oct-4 is on the chromatin is unclear. Indeed, it would be expected that Oct-4 may leave naturally its binding site to allow DNA replication during S phase. So, It would be interesting to check if Oct-4 is binding DNA in S phase? What happening to its phosphorylation state in S phase? How does it relate to the S229 phosphorylation?

The authors should cite the recent manuscript published by Huck Hui Ng and colleagues PMID: 26232226. How do their results relate to these mechanisms described in this publication?

In addition, the authors only look at few OCT-4 binding site in G2/M. Indeed, their data suggest that Oct-4 can still bind DNA during this phase of the cell cycle. It would be interesting to identify the genomic regions bound by Oct4 in G2/M and M/G1.

*Reviewer #2:* In this study, Shin et al. demonstrated that Oct4, a key regulator of pluripotent stem cells, is phosphorylated at S229 at G2/M phase but dephosphorylated at G1 phase of each cell cycle. The authors convincingly demonstrated that the Aurora kinase B and PP1, which play critical roles in kinetochore assembly and disassembly, regulate this modification. The authors argue that dephosphorylation of S229 at each G1 phase allows resetting the core transcriptional network. Overall, this study discovered a novel and interesting post-translational modification of Oct4, however, the biological consequence/relevance of this regulation is not established.

The main concerns are:

1) Data demonstrating that p-Oct4 dissociates from chromatin is not convincing. The main evidence supporting this point is Figure 1, where the authors performed ChIP-qPCR demonstrating that p-Oct4 cannot bind to Oct4 or Nanog promoter. However, because it is not clear whether the p-Oct4 antibody enables chromatin immunoprecipitation, the lack of p-Oct4 binding can also be explained as the antibody lacks ChIP capacity. Furthermore, if p-Oct4 cannot bind to chromatin and is poor in target gene transactivation, one should expect decreased expression of Oct4 target genes in ESCs treated with nocodazole (e.g. 10 hrs). Has this been observed?

2) Because suppression of PP1 by OKA affects activities of many other PP1 substrates, the observed differences in cell cycle progression, chromatin binding, and nascent mRNA transcription in Figure 4 and Figure 5 cannot be simply attributed to Oct4 S229 phosphorylation. It is known that PP1 activity is required for kinetochore disassembly and mitotic exit, which may explain the delayed G1 re-entry in Figure 4. Furthermore, PP1 activates Rb family proteins at G1 phase, which may directly contribute to the difference in nascent mRNA transcription in Figure 4 and Figure 5. Similarly, the ChIP-PCR data in Figure 4 and Figure 5 may also be explained by differences in cell cycle profile and phosphorylation status of certain Oct4 co-factors. These data failed to directly address the roles of p-Oct4. To better establish functional relevance of this modification, the authors may characterize Zhbtc4 cells expressing the S229A or S229D mutant.

3) In Figure 6, the cell cycle profile of ESCs expressing the F271A mutant is quite different from ESCs expressing the S229D mutant, which is unexpected according to the proposed model. Does the S229D mutant retain some level of transactivation activity? Is the F271A mutant dominant negative?

4) It is difficult to fully appreciate the functional relevance of S229 modification. It seems that WT Oct4 is better than both S229A and S229D mutants in self-renewal and/or reprogramming assays (Figure 6 and Figure 6). Why phosphorylation of Oct4 at S229 at G2/M is important? Is S229D more stable than S229A in mitosis? Is the S229A as good as WT Oct4 in activating target genes and supporting self-renewal? Are great majority or only a small fraction of Oct4 molecules phosphorylated by Aurkb at G2/M?

*Reviewer #3*:

In this manuscript, the authors investigated the phosphorylation sites on the core pluripotency factor Oct4 and associated one of the phosphorylation site S229 with Aurora B and PP1 which are key cell cycle kinases and phosphatase. Interestingly, the phosphorylation status on Oct4 S229 seems to modulate Oct4 chromatin binding activity. And the authors proposed that the dynamic phosphorylation of Oct4 is important for its ejection and rebinding before and after mitosis. Although the precise role of Oct4 ejection and rebinding within a cell cycle is not clear, the authors hinted that this could be important to maintaining Oct4 transcription and pluripotency.

Overall, this is an interesting manuscript which links cell cycle kinase/phosphatase with the controlling of the activity of the key pluripotency factor Oct4. The authors performed nice biochemical and functional experiment to investigate this novel phosphorylation site on Oct4. Although some of the experiments are more correlative, overall the experiments support their conclusions.

Specific comments:

1) Does the concentration of nocodazole used in this study arrest cells in G2 phase or M phase? Aurora B is known to be most active in M phase; besides, M phase chromatin condensation are known to cause transcription factors ejection from chromatin. Thus it is better to perform FACS staining for a mitotic marker, so that the authors may be able to precisely locate M phase as the time in the cell cycle when Oct4 gets phosphorylated.

2) Figure 1 only shows there is reduced Oct4 binding to chromatin in nocodazole treated cells. This data cannot lead to the conclusion that phosphorylated Oct4 has reduced chromatin binding in nocodazole.

3) Aurora B plays an important role in maintaining mitotic arrest caused by spindle disrupting agents, like nocodazole. In Figure 2—figure supplement 1, was the depletion or inhibition of Aurora B causing cells to exit from mitotic arrest, and therefore indirectly lead to the dephosphorylation of Oct4? Similarly, the use of OKA seems to enrich cells towards 4N, will this indirectly promote Oct4 phosphorylation?

4) If Aurora B kinase is activated exogenously during interphase, will this trigger premature Oct4 phosphorylation and ejection from chromatin during interphase?

5) The conclusion on Oct4 phosphorylation and chromatin ejection based on experiments using nocodazole treatment is only correlative. Authors only showed that during mitosis Oct4 gets phosphorylated by Aurora B which correlates with its cytoplasmic localization. However, the majority of transcription factors are ejected from chromatin during mitosis due to chromatin condensation. Thus, the ejection of Oct4 from chromatin during mitosis could be due to the physical condensation of chromatin rather than the phosphorylation by AuroraB.

6) Is S229A mutant more potent in binding to Oct4 target genes even with PP1 inhibition? Alternatively, will S229A mutant constitutively bind to chromatin even in M phase? If Oct4 S229A mutant is more potent is chromatin binding, then why does it seem to possess weaker activity and also caused a reduction in reprogramming efficiency?

---

## [Author Response]

*1) Please determine when in the cell cycle Oct4 is bound to DNA and perform CHIP-seq during distinct phases of the cell cycle (e.g., G2M vs G1) to define which regions are bound in a cell-cycle dependent manner.*

According to the reviewers’ comments, we mapped the genomewide occupancy of Oct4 at G2/M and G1 phase in E14 ESCs by ChIP-seq. We found high-confidence peaks (with p value<10−5) at G2/M (9204) and G1 (24548) phase (Figure 6 and [Supplementary-material SD3-data]). Expectedly, Oct4 bound much more regions of genome at G1 phase rather than G2/M phase (Figure 6). Furthermore, mean peak density of Oct4 bound regions was higher in G1 phase than G2/M phase (Figure 6) implicating that Oct4 genome-widely resets the target genes at G1 phase.

Next to identify reset region by Oct4, we discovered 9092 of genomic regions enriched by Oct4 more than two-fold increase in G1 phase compared to G2/M phase ([Supplementary-material SD4-data]). We categorized these as the Oct4 resetting peaks, which are significantly enriched in not only pluripotency but also cell cycle categories (Figure 6). For example, we showed that Oct4 more strongly binds to the regions of both Nanog and Sox2 at G1 phase rather than G2/M phase, supporting that Oct4 resets its target genes at G1 phase (Figure 6).

On the other hands, we identified G2/M or G1 specific peaks ([Supplementary-material SD3-data]). Majority of G2/M peaks, are included in G1 peaks (Figure 6) and show similar genomewide occupancy pattern compared with G1 peaks (Figure 6). However, there were several regions which are bound by Oct4 specifically at G2/M or G1 phase, implying that Oct4 may target genes in a cell cycle dependent manner. Our data might provide insights about cell cycle dependent transcriptional regulation of Oct4.

*2) Please provide evidence for the robustness of the pOct4 Ab for CHIP.*

The reviewers raise an important point. Throughout the Figure 1—figure supplement 2, we confirmed that phosphor-Oct4(S229) antibody specifically recognizes phosphorylated Oct4 (S229). To further address this question of the robustness of pOct4 Ab for ChIP, we first examined the capacity of this phospho-Oct4(S229) antibody to pull down its own pOct4(S229) by immunoprecipitation assay under the same condition as we performed ChIP experiment. As a result, the antibody successfully pulled-down p-Oct4(S229) in E14 mouse embryonic stem cells (Figure 1—figure supplement 2), indicating that this antibody can pull-down phosphor-Oct4(S229).

Despite p-Oct4(S229) being pulled down, the chromatin was not. Moreover, it has mentioned that steric and electrostatic clashes induced by phosphorylation at S229 residue can cause loss of DNA binding ability (Saxe et al., 2009). From above, weak ChIP signal of pOct4(S229) in Figure 1 is basically caused by weak binding of p-Oct4(S229) to chromatin, not by the quality of antibody.

3) Consider the criticism from Reviewer #2 regarding the lack of specificity of the OKA experiments.

To complement the experiment which may be misrepresented by cell cycle effect of OKA treatment, we performed ChIP-qPCR assay in Zhbtc4 ESCs stably expressing wild-type Oct4 (WT) and phsphor-defect mutant (S229A). When cells were released into G1 phase, OKA treatment significantly prevented the binding of Oct4(WT) to target genes. On the other hand, binding of Oct4(S229A) was relatively less affected even in treatment of OKA (Figure 4—figure supplement 1).

Even though Oct4(S229A) has the ability of binding to target genes, the reason why Oct4(S229A) has the low capacity of reprogramming can be explained by O-GlcNAcylation of Oct4. In our previous study, we identified that serine 228 O-GlcNAcylation of Oct4 is crucial for maintaining pluripotency (Jang et al., 2012). We found that Oct4 (S229A) mutation led to reduction of Oct4-O-GlcNAcylation which is critical for Oct4 activity, thereby S229A may reduce DNA binding of Oct4 and reprogramming efficiency. Based on these findings we concluded that dephosphorylation is paramount for the resetting of Oct4 to target genes during the M/G1 transition.

*4) Provide better characterization of the nocodozole effects as stressed by Reviewer #3.*

It is reported that during S phase Aurkb is activated then accumulates histone H3 serine 10 phosphorylation (H3S10p), a well-known mitotic marker, in mouse embryonic stem cells (Mallm and Rippe, 2015). In consistent with this report, when we released G2/M arrested E14 mouse embryonic stem cells, H3S10p levels retained highly through S-G2/M phase (7-10 hrs after release) (Figure 4—figure supplement 1), indicating that it is hard to distinguish G2 from M phase of stem cells with a mitotic marker (H3S10p). Next, to examine weather Oct4 binding to target genes is dependent on physical condensation of chromatin as reviewer commented in question #5, we chased p-Oct4 level and binding of Oct4 to target genes throughout the S-G2/M phase, (Figure 4—figure supplement 1). As a result, binding of Oct4 to target chromatin started to decline at S-G2/M phase (7-9 hr after release) and increased at the M/G1 transition (10hr after release) in parallel with the p-Oct4(S229) level. Based on this finding, we suggest that dissociation of phosphorylated Oct4 from chromatin proceeds to mitotic chromatin condensation. Nonetheless, we agree with the reviewer’s opinion that M phase chromatin condensation partly contributes to the ejection of Oct4 from chromatin.

Reviewer #1:

*The precise timing by which Oct-4 is on the chromatin is unclear. Indeed, it would be expected that Oct-4 may leave naturally its binding site to allow DNA replication during S phase. So, It would be interesting to check if Oct-4 is binding DNA in S phase? What happening to its phosphorylation state in S phase? How does it relate to the S229 phosphorylation?*

We previously showed that p-Oct4(S229) begin to be detected in the S phase by western blot and FACS analysis (Figure 1 and Figure 4). According to reviewer’s comments, we performed ChIP-qPCR to examine Oct4 binding to chromatin in S phase (6 hr after release) as the same condition as in Figure 4 (Figure 9, see below). We found that binding of Oct4 to target genes was slightly reduced in the S phase, implying that phosphorylated Oct4 in S phase might begin to be dissociated from chromatin. At this point, we cannot know how much Oct4 phosphorylation is contributed to Oct4 dissociation during S phase because many DNA binding proteins are believed to be dissociated from chromatin during replication. Therefore, at this stage, we cannot exclude the possibility of physical dissociation of Oct4 during replication. We will consider more detailed mechanism in near future.

Author response image 1.Oct4 bound weakly to target genes in the S phase.**DOI:**
http://dx.doi.org/10.7554/eLife.10877.022

*The authors should cite the recent manuscript published by Huck Hui Ng and colleagues PMID: 26232226. How do their results relate to these mechanisms described in this publication?*

We thank the reviewer for this suggestion. The study revealed that S and G2 phases are gatekeepers of differentiation in human embryonic stem cells. We cited the paper in the Introduction and Discussion section.

*In addition, the authors only look at few OCT-4 binding site in G2/M. Indeed, their data suggest that Oct-4 can still bind DNA during this phase of the cell cycle. It would be interesting to identify the genomic regions bound by Oct4 in G2/M and M/G1.*

According to the reviewer’s comments, we mapped the genomewide occupancy of Oct4 at G2/M and G1 phase in E14 ESCs by ChIP-seq. We found high-confidence peaks (with p value<10−5). at G2/M (9204) and G1 (24548) phase (Figure 6). Expectedly, Oct4 bound much more regions of genome at G1 phase rather than G2/M phase (Figure 6). Furthermore, mean peak density of Oct4 bound regions was higher in G1 phase than G2/M phase (Figure 6) implicating that Oct4 genome-widely resets the target genes at G1 phase.

Next to identify reset region by Oct4, we discovered 9092 of genomic regions enriched by Oct4 more than two-fold increase in G1 phase compared to G2/M phase ([Supplementary-material SD4-data]). We categorized these as the Oct4 resetting peaks, which are significantly enriched in not only pluripotency but also cell cycle categories (Figure 6). For example, we showed that Oct4 more strongly binds to the regions of both Nanog and Sox2 at G1 phase rather than G2/M phase, supporting that Oct4 resets its target genes at G1 phase (Figure 6).

On the other hands, we identified G2/M or G1 specific peaks ([Supplementary-material SD3-data]). Majority of G2/M peaks, are included in G1 peaks (Figure 6) and show similar genomewide occupancy pattern compared with G1 peaks (Figure 6). Interestingly, there were several regions which are bound by Oct4 specifically at G2/M or G1 phase, implying that Oct4 may target genes in a cell cycle dependent manner. Our data might provide insights about cell cycle dependent transcriptional regulation of Oct4.

Reviewer #2:

*1) Data demonstrating that p-Oct4 dissociates from chromatin is not convincing. The main evidence supporting this point is Figure 1, where the authors performed ChIP-qPCR demonstrating that p-Oct4 cannot bind to Oct4 or Nanog promoter. However, because it is not clear whether the p-Oct4 antibody enables chromatin immunoprecipitation, the lack of p-Oct4 binding can also be explained as the antibody lacks ChIP capacity. Furthermore, if p-Oct4 cannot bind to chromatin and is poor in target gene transactivation, one should expect decreased expression of Oct4 target genes in ESCs treated with nocodazole (e.g. 10 hrs). Has this been observed?*

Throughout Figure 1—figure supplement 2, we confirmed that phosphor-Oct4(S229) antibody specifically recognizes phosphorylated Oct4 (S229). To further address this question of the robustness of pOct4 Ab for ChIP, we first examined the capacity of this phospho-Oct4(S229) antibody to pull down its own pOct4(S229) by immunoprecipitation assay under the same condition as we performed ChIP experiment. As a result, the antibody successfully pulled-down p-Oct4(S229) in E14 mouse embryonic stem cells (Figure 1—figure supplement 2), indicating that this antibody can pull-down phosphor-Oct4(S229).

Despite p-Oct4(S229) protein being pulled down, the chromatin was not. Moreover, it has mentioned that steric and electrostatic clashes induced by phosphorylation at S229 residue can cause loss of DNA binding ability (Saxe et al., 2009). From above, weak ChIP signal of pOct4(S229) in Figure 1 is basically caused by weak binding of p-Oct4(S229) to chromatin, not by the quality of antibody.

*2) Because suppression of PP1 by OKA affects activities of many other PP1 substrates, the observed differences in cell cycle progression, chromatin binding, and nascent mRNA transcription in Figure 4 and Figure 5 cannot be simply attributed to Oct4 S229 phosphorylation. It is known that PP1 activity is required for kinetochore disassembly and mitotic exit, which may explain the delayed G1 re-entry in Figure 4. Furthermore, PP1 activates Rb family proteins at G1 phase, which may directly contribute to the difference in nascent mRNA transcription in Figure 4 and Figure 5. Similarly, the ChIP-PCR data in Figure 4 and Figure 5 may also be explained by differences in cell cycle profile and phosphorylation status of certain Oct4 co-factors. These data failed to directly address the roles of p-Oct4. To better establish functional relevance of this modification, the authors may characterize Zhbtc4 cells expressing the S229A or S229D mutant.*

At the reviewer’s suggestion, we performed ChIP-qPCR assay in Zhbtc4 ESCs stably expressing wild-type Oct4(WT) and phsphor-defect mutant(S229A). When cells were released into G1 phase, OKA treatment significantly prevented the binding of Oct4(WT) to target genes. On the other hand, binding of Oct4(S229A) was relatively less affected even in treatment of OKA (Figure 4—figure supplement 1).

Even though Oct4(S229A) has the ability of binding to target genes, the reason why Oct4(S229A) has the low capacity of reprogramming can be explained by O-GlcNAcylation of Oct4. In our previous study, we identified that serine 228 O-GlcNAcylation of Oct4 is crucial for maintaining pluripotency (Jang et al., 2012). We found that Oct4 (S229A) mutation led to reduction of Oct4-O-GlcNAcylation which is critical for Oct4 activity, thereby S229A may reduce DNA binding of Oct4 and reprogramming efficiency. Based on these findings we concluded that dephosphorylation is paramount for the resetting of Oct4 to target genes during the M/G1 transition.

3) In Figure 6, the cell cycle profile of ESCs expressing the F271A mutant is quite different from ESCs expressing the S229D mutant…

As the reviewer commented, F271A mutation disrupts pluripotency and cell cycle of mouse embryonic stem cells more severely than S229D (Figure 7). Even though we still do not understand the mechanism clearly, we think that PP1-binding to Oct4 may have additional function for resetting transcription at the next G1 entrance, probably by affecting the epigenetic environments around Oct4 binding regions. We will be happy if you give us a chance to solve the detailed mechanism in next time.

*… which is unexpected according to the proposed model. Does the S229D mutant retain some level of transactivation activity?*

We have already published that S229D mutant has lower activity (around 25% ) in comparison with the activity of Oct4 (WT) (Jang et al., 2012).

*Is the F271A mutant dominant negative?*

In Figure 7, Zhbtc4 cells harboring F271A mutant without doxycycline seems to be normal. However, once endogenous Oct4 was depleted in these cells by treatment of dox, cell cycle profile in F271A cells were retarded (Figure 7) and lost the AP staining activity (Figure 7). From above, we believe that F271A mutant does not seem to be dominant-negative.

*4) It is difficult to fully appreciate the functional relevance of S229 modification. It seems that WT Oct4 is better than both S229A and S229D mutants in self-renewal and/or reprogramming assays (Figure 6 and Figure 6). Why phosphorylation of Oct4 at S229 at G2/M is important?*

We believe that the cycle of phosphorylation and dephosphorylation of Oct4 during cell cycle is important. Throughout this study, we revealed that the resetting of Oct4 occurs rapidly during the exit from mitosis to prevent postmitotic cell fate changes in ESCs. In addition, we suggested that the potential of ESCs to differentiate might be derived from the small window of the M/G1 transition and phosphorylation of Oct4 is deeply involved in this mechanism.

*Is S229D more stable than S229A in mitosis?*

We treated cycloheximide to Zhbtc4 embryonic stem cells stably expressing each Oct4 WT and mutant form and assessed by western blot (Figure 10). We couldn’t find any significant differences between Oct4 WT, S229A and S229D. We suggest that phosphorylation of Oct4 is not crucial factor for determining protein stability.

Author response image 2.Serine 229 mutation didn’t affect Oct4 stability.**DOI:**
http://dx.doi.org/10.7554/eLife.10877.023

*Is the S229A as good as WT Oct4 in activating target genes and supporting self-renewal?*

To address this question, we performed ChIP-qPCR assay in Zhbtc4 ESCs stably expressing wild-type Oct4(WT) and phsphor-defect mutant(S229A). When cells were released into G1 phase, OKA treatment significantly prevented the binding of Oct4(WT) to target genes. On the other hand, binding of Oct4(S229A) was relatively less affected even in treatment of OKA (Figure 4—figure supplement 1). Even though Oct4(S229A) has the ability of binding to target genes, the reason why Oct4(S229A) has the low capacity of reprogramming can be explained by O-GlcNacylation of Oct4. In our previous study, we revealed that serine 228 O-GlcNacylation of Oct4 is crucial for maintaining pluripotency (Jang et al., 2012). We found that Oct4 (S229A) mutation led to reduction of Oct4-O-GlcNAcylation which is critical for Oct4 activity, thereby S229A may reduce DNA binding of Oct4 and reprogramming efficiency.

*Are great majority or only a small fraction of Oct4 molecules phosphorylated by Aurkb at G2/M?*

At this point, we showed that large portion of phosphorylated Oct4 (S229) was detected in cytoplasm at G2/M phase (Figure 1).

Reviewer #3:

*1) Does the concentration of nocodazole used in this study arrest cells in G2 phase or M phase? Aurora B is known to be most active in M phase; besides, M phase chromatin condensation are known to cause transcription factors ejection from chromatin. Thus it is better to perform FACS staining for a mitotic marker, so that the authors may be able to precisely locate M phase as the time in the cell cycle when Oct4 gets phosphorylated.*

It is reported that during S phase Aurkb is activated then accumulates histone H3 serine 10 phosphorylation (H3S10p), a well-known mitotic marker, in mouse embryonic stem cells (Mallm and Rippe, 2015). In consistent with this report, when we released G2/M arrested E14 mouse embryonic stem cells, H3S10p levels retained highly through S-G2/M phase (7-10 hrs after release) (Figure 4—figure supplement 1), indicating that it is hard to distinguish G2 from M phase of stem cells with a mitotic marker (H3S10p). Next, to examine weather Oct4 binding to target genes is dependent on physical condensation of chromatin as reviewer commented in question #5, we chased p-Oct4 level and binding of Oct4 to target genes throughout the S-G2/M phase, (Figure 4—figure supplement A and B). As a result, binding of Oct4 to target chromatin started to decline at S-G2/M phase (7-9 hr after release) and increased at the M/G1 transition (10hr after release) in parallel with the p-Oct4(S229) level. Based on this finding, we suggest that dissociation of phosphorylated Oct4 from chromatin proceeds to mitotic chromatin condensation. Nonetheless, we agree with the reviewer’s opinion that M phase chromatin condensation partly contributes to the ejection of Oct4 from chromatin.

*2) Figure 1 only shows there is reduced Oct4 binding to chromatin in nocodazole treated cells. This data cannot lead to the conclusion that phosphorylated Oct4 has reduced chromatin binding in nocodazole.*

To address this question, we mapped the genomewide occupancy of Oct4 at G2/M (nocodazole-treated cells) and G1 phase (release of nocodazole-treated cells) in E14 ESCs by ChIP-seq. We found that Oct4 bound reduced regions of genome at G2/M phase, compared to G1 phase (Figure 6). Furthermore, mean peak density of Oct4 bound regions was lower in G2/M phase than G1 phase (Figure 6). These findings imply that at G2/M phase, Oct4 dissociates genome-widely from chromatin, supporting our ChIP data in Figure 1.

*3) Aurora B plays an important role in maintaining mitotic arrest caused by spindle disrupting agents, like nocodazole. In Figure 2—figure supplement 1, was the depletion or inhibition of Aurora B causing cells to exit from mitotic arrest, and therefore indirectly lead to the dephosphorylation of Oct4?*

The reviewer raises an important point. Indeed, inhibition of cell cycle related proteins such as Aurkb can result in cell cycle arrest. Consequently, we analyzed cell cycle of E14 mouse embryonic stem cells treated with Aurkb inhibitor under the same conditions applied in Figure 2. As a result, we confirmed that the treated time and concentration of inhibitor we used did not alter the cell cycle profile compared to G2/M arrested control cells. This data was added to Figure 2 of the revised manuscript.

*Similarly, the use of OKA seems to enrich cells towards 4N, will this indirectly promote Oct4 phosphorylation?*

When G2/M arrested cells were released into normal serum, OKA treatment retarded re-entry into the next G1 phase (Figure 4). In spite of delayed cell cycle, as time pass by the portion of cells remaining at G2/M phase was decreased gradually (Figure 4). However, p-Oct4(S229) levels highly retained even though 2 hours of releasing. In this study, we showed that Aurkb directly phosphorylates Oct4(S229) and PP1 directly dephosphorylates Oct4(S229), therefore, implying that OKA treatment can accumulate the phosphorylation of Oct4 by blocking the PP1-mediated dephosphorylation of Oct4.

*4) If Aurora B kinase is activated exogenously during interphase, will this trigger premature Oct4 phosphorylation and ejection from chromatin during interphase?*

It has been reported that Aurkb is tightly regulated in a cell cycle dependent manner (Carmena et al., 2012). Usually, Aurkb in cancer cells is activated during G2/M phase by INCENP-hespin kinase. In addition, in mouse embryonic stem cells, Aurkb is reported to be active during S-G2/M phase (Mallm and Rippe, 2015). We also showed that Aurkb begin to phosphorylate Oct4 at S phase (Figure 4). In addition, to activate Aurkb completely, Aurkb should binds and phosphorylates to INCENP (Carmena et al., 2012), implying that constitutively-active Aurkb does not exist. Therefore, we do not believe that Aurkb can be activated during G1 interphase. Thus, we showed that Oct4(S229D), a mimic of Aurkb-phosphorylating Oct4, retards the resetting at G1 phase (Figure 7), which hopefully answers the reviewer’s question.

*5) The conclusion on Oct4 phosphorylation and chromatin ejection based on experiments using nocodazole treatment is only correlative. Authors only showed that during mitosis Oct4 gets phosphorylated by Aurora B which correlates with its cytoplasmic localization. However, the majority of transcription factors are ejected from chromatin during mitosis due to chromatin condensation. Thus, the ejection of Oct4 from chromatin during mitosis could be due to the physical condensation of chromatin rather than the phosphorylation by AuroraB.*

We thank the reviewer for pointing out this concern. We found several lines of evidence implying that dissociation of phosphorylated Oct4 from chromatin proceeds to mitotic chromatin condensation and explained detailed descriptions in response to question #1. We agree with the reviewer’s opinion that M phase chromatin condensation partly contributes the ejection of Oct4 from chromatin.

*6) Is S229A mutant more potent in binding to Oct4 target genes even with PP1 inhibition? Alternatively, will S229A mutant constitutively bind to chromatin even in M phase? If Oct4 S229A mutant is more potent is chromatin binding, then why does it seem to possess weaker activity and also caused a reduction in reprogramming efficiency?*

According to the reviewer’s comment’s, we compared DNA binding ability of Oct4(S229A) to Oct4(WT). Oct4(S229A) binding is slightly higher than Oct4(WT) at G2/M phase and G1 phase with OKA treatment. While, Oct4(S229A) have lower reporter activity in NIH-3t3 cells (Figure 7—figure supplement 1) and reprogramming efficiency (Figure 7) than Oct4(WT). This effect can be explained with O-GlcNacylation of Oct4. In our previous study about significance of serine 228 O-GlcNacylation of Oct4 to maintaining pluripotency (Jang et al., 2012), we found that Oct4 (S229A) mutation led to significant reduction of Oct4-O-GlcNAcylation which is critical for Oct4 activity, thereby S229A may reduce DNA binding of Oct4 and reprogramming efficiency.

References

Carmena, M., Wheelock, M., Funabiki, H., and Earnshaw, W.C. (2012). The chromosomal passenger complex (CPC): from easy rider to the godfather of mitosis. Nature reviews. Molecular Cell Biology 13, 789-803. doi:10.1038/nrm3474

Jang, H., Kim, T.W., Yoon, S., Choi, S.Y., Kang, T.W., Kim, S.Y., Kwon, Y.W., Cho, E.J., and Youn, H.D. (2012). O-GlcNAc regulates pluripotency and reprogramming by directly acting on core components of the pluripotency network. Cell Stem Cell 11, 62-74. doi:10.1016/j.stem.2012.03.001

Mallm, J.P., and Rippe, K. (2015). Aurora Kinase B Regulates Telomerase Activity via a Centromeric RNA in Stem Cells. Cell Rep 11, 1667-1678. doi:10.1016/j.celrep.2015.05.015

Saxe, J.P., Tomilin, A., Scholer, H.R., Plath, K., and Huang, J. (2009). Post-translational regulation of Oct4 transcriptional activity. PLOS ONE 4, e4467. doi:10.1371/journal.pone.0004467